# PCH1 integrates circadian and light-signaling pathways to control photoperiod-responsive growth in Arabidopsis

He Huang[1], Chan Yul Yoo[2], Rebecca Bindbeutel[1], Jessica Goldsworthy[3], Allison Tielking[4], Sophie Alvarez[1†], Michael J Naldrett[1†], Bradley S Evans[1], Meng Chen[2], Dmitri A Nusinow[1]*

[1]Donald Danforth Plant Science Center, St. Louis, United States; [2]Department of Botany and Plant Sciences, Institute of Integrative Genome Biology, University of California at Riverside, Riverside, United States; [3]Michigan State University, East Lansing, United States; [4]Mary Institute and Saint Louis Country Day School, St. Louis, United States

**Abstract** Plants react to seasonal change in day length through altering physiology and development. Factors that function to harmonize growth with photoperiod are poorly understood. Here we characterize a new protein that associates with both circadian clock and photoreceptor components, named PHOTOPERIODIC CONTROL OF HYPOCOTYL1 (PCH1). *pch1* seedlings have overly elongated hypocotyls specifically under short days while constitutive expression of *PCH1* shortens hypocotyls independent of day length. PCH1 peaks at dusk, binds phytochrome B (phyB) in a red light-dependent manner, and co-localizes with phyB into photobodies. PCH1 is necessary and sufficient to promote the biogenesis of large photobodies to maintain an active phyB pool after light exposure, potentiating red-light signaling and prolonging memory of prior illumination. Manipulating PCH1 alters PHYTOCHROME INTERACTING FACTOR 4 levels and regulates light-responsive gene expression. Thus, PCH1 is a new factor that regulates photoperiod-responsive growth by integrating the clock with light perception pathways through modulating daily phyB-signaling.

**\*For correspondence:** meter@ danforthcenter.org

**Present address:** †Proteomics and Metabolomics Facility, Center for Biotechnology, University of Nebraska-Lincoln, Lincoln, United States

**Competing interests:** The author declares that no competing interests exist.

## Introduction

Plants have evolved to coordinate physiology and phenology with seasonal variation in the environment (*Wilczek et al., 2010*). These adaptations to changing day length are called photoperiodic responses, which are regulated by both the circadian clock and specific signaling pathways, including light sensory systems (*Shim and Imaizumi, 2015*). In plants, photoperiod regulates myriad processes, including the transition to flowering (*Valverde et al., 2004*), cold acclimation (*Lee and Thomashow, 2012*), and growth (*Niwa et al., 2009*; *Nomoto et al., 2012*). In *Arabidopsis*, daily hypocotyl elongation is accelerated in short days compared to long day conditions, and requires both the circadian clock and light signals to properly react to changing photoperiods (*Niwa et al., 2009*; *Nozue et al., 2007*).

Circadian clocks provide an adaptive advantage by synchronizing internal physiology to the external environment, allowing for an efficient allocation of resources in plants (*Dodd et al., 2005*). More than 20 clock components have been characterized in *Arabidopsis*, forming a complex network of interlocking transcription-translation feedback loops (*Hsu and Harmer, 2014*; *Nagel and Kay,*

**eLife digest** Most living things possess an internal "circadian" clock that synchronizes many behaviors, such as eating, resting or growing, with the day-night cycle. With the help of proteins that can detect light, known as photoreceptors, the clock also coordinates these behaviors as the number of daylight hours changes during the year. However, it is not known how the clock and photoreceptors are able to work together.

The circadian clocks of animals and plants have evolved separately and use different proteins. In plants, a photoreceptor called phytochrome B responds to red light and regulates the ability of plants to grow. Most plants harness sunlight during the day, but grow fastest in the dark just before dawn. In 2015, researchers identified a new protein in a plant called Arabidopsis that is associated with several plant clock proteins and photoreceptors, including phytochrome B. However, the role of this new protein was not clear.

Now, Huang et al. – including many of the researchers from the 2015 work – studied the new protein, named PCH1, in more detail. The experiments show that PCH1 is a critical link that regulates the daily growth of Arabidopsis plants in response to the number of daylight hours. PCH1 stabilizes the structure of phytochrome B so that it remains active, even in the dark. This prolonged activity acts as a molecular memory of prior exposure to light and helps to prevent plants from growing too much in the winter when there are fewer hours of daylight. Since PCH1 is also found in other species of plants, it may play the same role in regulating growth of major crop plants.

The next challenge is to understand how the binding of PCH1 to phytochrome B alters the photoreceptor's activity. In the future, Huang et al. hope to find out if manipulating the activity of PCH1 can improve the growth of crops in places where there is a large change in day length across the seasons.

*2012*; *Pokhilko et al., 2012*). Among them, a tripartite protein complex named the Evening Complex (EC) regulates circadian rhythms and suppresses hypocotyl growth in the evening (*Nusinow et al., 2011*). Mutations in any of the EC components, *EARLY FLOWERING 3 (ELF3)* (*Hicks et al., 2001*), *EARLY FLOWERING 4 (ELF4)* (*Doyle et al., 2002*) or *LUX ARRHYTHMO (LUX)* (*Hazen et al., 2005*; *Onai and Ishiura, 2005*), leads to arrhythmic circadian oscillations, elongated hypocotyls, and early flowering regardless of day length (*Nagel and Kay, 2012*). The EC regulates hypocotyl elongation by repressing the expression of two critical bHLH transcription factors *PHYTOCHROME INTERACTING FACTOR 4* and *5 (PIF4* and *PIF5)* (*Nusinow et al., 2011*), which are two key regulators in phytochrome-mediated light signaling pathways (*Huq and Quail, 2002*; *Khanna et al., 2004*). Furthermore, ELF3 directly binds to the red light photoreceptor phytochrome B (phyB) (*Liu et al., 2001*) and the E3-ligase CONSTITUTIVE PHOTOMORPHOGENIC 1 (COP1) (*Yu et al., 2008*), connecting the clock to light signaling.

Arabidopsis possesses five red/far-red light absorbing phytochromes (phyA to E) (*Clack et al., 1994*; *Sharrock and Quail, 1989*). Phytochromes are converted to the Pfr (active) form upon red (660 nm) light treatment, and reverted to the Pr (inactive) form either upon far-red (730 nm) light exposure or by incubation in the dark in a process termed dark reversion (*Rockwell et al., 2006*). Signaling through phytochromes regulates germination, shade avoidance, circadian rhythms, photosynthesis, hypocotyl growth and flowering time (*Kami et al., 2010*). During the day, phytochromes play a prominent role sensing environmental light signals to suppress growth: phyB in the Pfr state binds to PIFs (such as PIF3, 4 and 5) to regulate their post-translational turnover (*Bauer et al., 2004*; *Lorrain et al., 2008*; *Nozue et al., 2007*). Taken together, daily growth rhythms in seedlings are the result of both post-translational degradation of PIF3, 4, and 5 by phytochromes (*Lorrain et al., 2008*; *Soy et al., 2012*) and transcriptional regulation of *PIF4* and *PIF5* by the EC (*Nozue et al., 2007*; *Nusinow et al., 2011*).

Photoconversion of phyB by red light induces its localization to discrete subnuclear domains named photobodies (*Chen and Chory, 2011*; *Chen et al., 2003*). Light conditions that drive the Pr/Pfr equilibrium towards Pfr will promote formation of large photobodies in vivo (*Chen et al., 2003*), which correlates with the photoinhibition of hypocotyl elongation and the degradation of PIF3

(*Chen et al., 2003*; *Van Buskirk et al., 2014*). Since proper degradation of PIFs is critical to regulate growth (*Al-Sady et al., 2006*; *Lorrain et al., 2008*), one proposed function of photobodies is to stabilize the phyB Pfr form, which allows active phyB to continue controlling the level of PIFs and suppressing hypocotyl growth in prolonged darkness or in short-days (*Rausenberger et al., 2010*; *Van Buskirk et al., 2014*). Current mathematical models of red-light signaling dynamics predict a yet undiscovered factor that directly modulates photobody formation in vivo in response to light (*Klose et al., 2015*).

Here we present the characterization of an EC-associated protein called PCH1 (for <u>P</u>HOTOPERI-<u>O</u>DIC <u>C</u>ONTROL OF <u>H</u>YPOCOTYL 1). Our results define PCH1 as a new clock-regulated phytochrome-binding factor that regulates photoperiodic growth by stabilizing phyB-containing photobodies in the evening, thereby providing a molecular mechanism for prolonging red-light signaling after prior light exposure.

## Results

### PCH1 (*At2g16365.2*) encodes a conserved, evening-peaked, EC-associated protein

A protein encoded by *At2g16365,* a gene that was described as required for transcriptional responses to lincomycin-induced chloroplast damage (*Ruckle et al., 2012*), was repeatedly co-purified with the EC by tandem affinity-purification coupled with mass spectrometry (AP-MS) analyses (*Huang et al., 2015*). According to the TAIR10 database (*Lamesch et al., 2011*), *At2g16365* has four splice variants encoding different protein products, three of which contain an F-box domain (*At2g16365.1, 3 and 4*) (*Figure 1—figure supplement 1A*). All peptides from AT2G16365 that co-purified with the EC were mapped to *At2g16365.2* (*Figure 1—supplement 2* and *Huang et al., 2015*), which contains the first two exons and lacks the sequence encoding the F-box domain. Semi-quantitative RT-PCR analysis and RNA-seq reads from a publically available RNA-seq dataset (*Gulledge et al., 2014*) indicated that only *At2g16365.2* is transcribed (*Figure 1—figure supplement 1B* and *Figure 1—figure supplement 3*). Therefore, all presented constructs are based on the dominant isoform *At2g16365.2*. Furthermore, a T-DNA insertion loss-of-function line in *At2g16365* (SALK_024229, *Ruckle et al., 2012*) resulted in a short-day-specific hypocotyl phenotype (described below). Thus, the *At2g16365* gene was renamed <u>P</u>HOTOPERIODIC <u>C</u>ONTROL OF <u>H</u>YPOCOTYL 1 (*PCH1*).

Examination of microarray data from long day, 12L:12D and short day time courses (Light: Dark hours = 16:8, 12:12 and 8:16, respectively) from the DIURNAL database (*Michael et al., 2008b*; *Mockler et al., 2007*) found that expression of *PCH1* cycled with a peak of expression occurring in the evening. *PCH1* mRNA accumulates after dawn, reaches a maximum at Zeitgeber time 8 (ZT8) and decreases towards the end of night (*Figure 1A*). This expression pattern was validated by quantitative PCR (qPCR) analyses using cDNA samples of short-day grown seedlings (*Figure 1B*).

To test if PCH1 protein levels oscillate, a transgene expressing PCH1-His$_6$-FLAG$_3$ under the control of the *PCH1* promoter in *pch1* (SALK_024229) (PCH1p::PCH1) was generated and protein abundance was monitored during a short day time course. The PCH1-His$_6$-FLAG$_3$ fusion protein accumulates after dawn, reaches a peak level in the early evening (ZT9) and a trough at subjective dawn (*Figure 1C*), similar to *PCH1* expression (*Figure 1B*). Under 12L:12D and long day conditions, PCH1-His$_6$-FLAG$_3$ in PCH1p::PCH1 plants continues to peak near the dusk transition (*Figure 1—figure supplement 4*). In a *PCH1* constitutive expression line (PCH1ox3), PCH1 protein levels were constant under all photoperiods, unlike PCH1p::PCH1 (*Figure 1C* and *Figure 1—figure supplement 4*).

To determine if PCH1 is present in other plant species, PCH1 orthologs were identified. Pairwise alignments of Arabidopsis PCH1 with orthologs from *Oryza sativa, Brachypodium distachyon*, and *Populus trichocarpa* indicate percent identity is 19.6%, 20.81% and 32.03%, respectively (Clustal Omega, http://www.ebi.ac.uk/Tools/msa/clustalo/). The last 43 amino acids of the C-terminus are highly conserved among PCH1 orthologs (*Figure 1—figure supplement 5*). In addition, *PCH1* orthologs share the evening-phased expression pattern under 12L:12D (*Figure 1D*), suggesting that PCH1 may have conserved time-of-day-specific functions (*Michael et al., 2008a*).

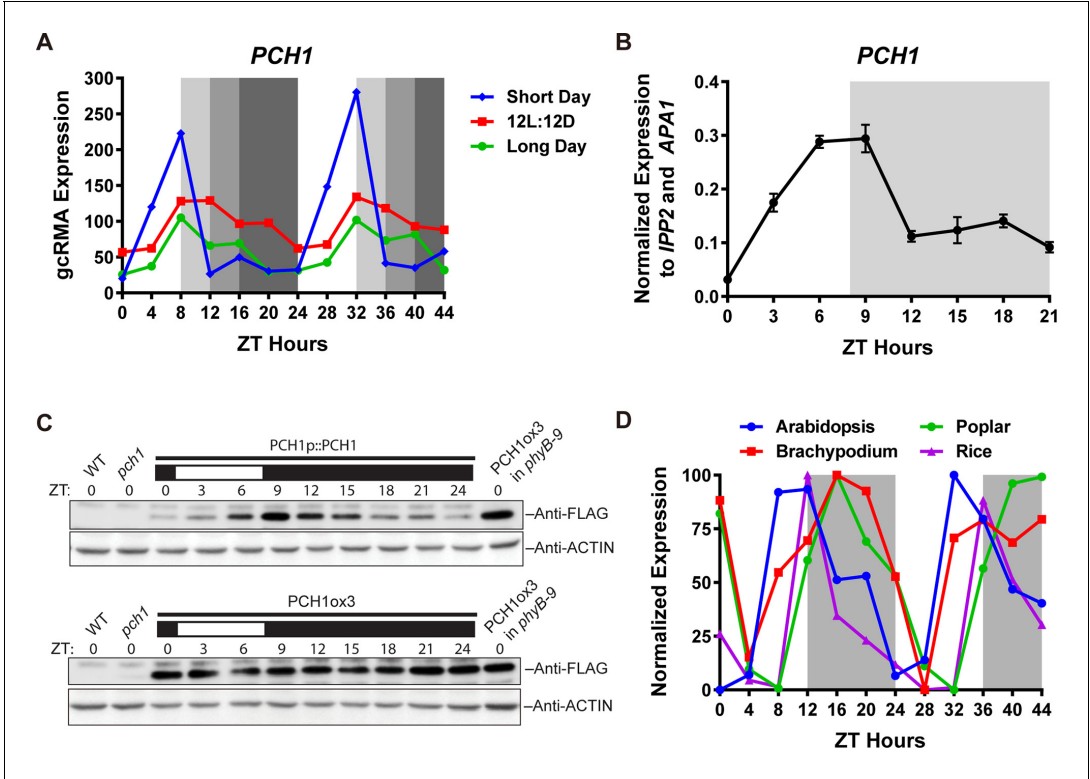

**Figure 1.** *PCH1 (At2g16365.2)* encodes a conserved evening-phased protein. (**A**) Time-course gcRMA (GeneChip Robust Multiarray Averaging) values of *PCH1* expression (from Diurnal database, http://diurnal.mocklerlab.org/, *Mockler et al., 2007*) under short day, 12L:12D and long day conditions (Light: Dark hours = 8:16, 12:12 and 16:8, respectively). Grey shading indicates dark period. (**B**) Time-course qPCR analysis of *PCH1* expression using cDNA samples (from ZT 0 to 24, with 3 hr intervals) of 4-day-old seedlings grown under short day conditions, normalized to *IPP2* and *APA1*. Mean ± SD (n=3 biological reps). (**C**) Anti-FLAG immunoblotting detecting PCH1-His$_6$-FLAG$_3$ levels using protein extracts from time-course samples (from ZT 0 to 24, with 3 hr intervals) of 4-day-old, short-day-grown PCH1p::PCH1 and PCH1ox3 plants, which express the tagged PCH1 protein driven by the PCH1 native promoter or the 35S CaMV promoter, respectively. Actin was used for normalization. Rectangles above blots represent light/dark conditions under which samples were flash frozen in liquid N$_2$, white = light and black = dark. Wild type (WT), *pch1* and PCH1ox3 in *phyB-9* were controls for immunoblots. (**D**) Normalized gcRMA values of PCH1 orthologs from *Arabidopsis thaliana (At2g16365), Brachypodium distachyon (Bradi2g46850), Oryza sativa* (Rice, *LOC_Os01g49310*), and *Populus trichocarpa* (Poplar, *POPTR_0004s16430.1*) under 12L:12D conditions from Diurnal database, http://diurnal.mocklerlab.org/, *Mockler et al., 2007*). Expression is normalized to min and max value.

The following figure supplements are available for figure 1:

**Figure supplement 1.** *At2g16365.2* is the predominant transcript of *PCH1*.

**Figure supplement 2.** Peptides identified by ELF3/4 AP-MS only mapped to the protein encoded by *At2g16365.2*.

**Figure supplement 3.** Available RNAseq data suggest only *At2g16365.2* is expressed.

**Figure supplement 4.** PCH1 levels peak at dusk under 12L:12D or long day conditions.

**Figure supplement 5.** Multiple sequence alignments of PCH1 orthologs at the C-terminus.

## *pch1* exhibits day-length specific defects in hypocotyl elongation

The association of PCH1 with the EC and light signaling components suggested that PCH1 may regulate hypocotyl elongation (*Huang et al., 2015*). Therefore, hypocotyl lengths of 4-day-old wild type, *phyB-9* (*Reed et al., 1993*), *elf4-2, elf3-2* (*Nusinow et al., 2011*) and *pch1* loss-of-function mutant (*Ruckle et al., 2012*) seedlings were compared under long day, 12L:12D and short day conditions (*Figure 2A*). Unlike *phyB-9, elf3-2* and *elf4-2*, which exhibit longer hypocotyls than wild type

under all photoperiods tested, *pch1* shows a day-length-specific defect (*Figure 2A*). Under long days, hypocotyls of *pch1* are not longer than wild type (*Figure 2A*). As the dark period extended to 12 hr, *pch1* exhibited a slightly but significantly longer hypocotyl than wild type (p<0.01) (*Figure 2A*). Under short days, *pch1* mutants elongated hypocotyls even further (p<0.0001) (*Figure 2A*). Constitutive expression of *PCH1* resulted in the opposite phenotype: PCH1ox3 mutants have shorter hypocotyls than wild type under all photoperiods (p<0.0001) (*Figure 2A*). PCH1p:: PCH1 rescued the hypocotyl length phenotype of *pch1* mutants in two independent lines (*Figure 2— figure supplement 1*), showing that the level and/or timing of *PCH1* expression is critical for proper regulation of hypocotyl elongation.

The evening-phased expression of *PCH1* suggested it may function to regulate growth rhythms at a specific time of day. Therefore, hypocotyl growth rates of wild type, *pch1*, PCH1ox3 and PCH1p::PCH1 were measured in short days by time-lapse imaging. Wild type plants showed rhythmic hypocotyl growth under short days, with a maximal growth rate at dawn, as described (*Nozue et al., 2007*) (*Figure 2B*). On the third night post-germination (from ZT56 to ZT72), *pch1* seedlings had higher hypocotyl growth rates than wild type during the night, especially during the $3^{rd}$ to the $5^{th}$ night (*Figure 2B*). Supporting the hypothesis that PCH1 is a suppressor of hypocotyl elongation, constitutive expression of *PCH1* in PCH1ox3 inhibited hypocotyl elongation throughout the night, while PCH1p::PCH1 restored the growth rate to wild type levels (*Figure 2B*).

## PCH1 is not required for circadian rhythms or sensitivity of flowering to photoperiod

Light signaling and the EC are critical for circadian rhythmicity and flowering pathways (*Nagel and Kay, 2012*; *Shim and Imaizumi, 2015*), therefore the role of PCH1 in circadian rhythms and time to flowering was investigated. To determine if *PCH1* regulates the circadian oscillator, a *CCA1*-promoter driven *LUCIFERASE (CCA1::LUC)* was used to monitor endogenous rhythms (*Pruneda-Paz et al., 2009*). The luciferase activity of *CCA1::LUC* in wild type, *pch1* and PCH1ox3 oscillates with a period of ~24 hr (23.20 ± 0.45, 23.13 ± 0.37, and 22.97 ± 0.32 hr, respectively, mean ± SD, n = 8) (*Figure 3A,B* and *Figure 3—figure supplement 1*), showing that *PCH1* levels do not affect the circadian period of the reporter. However, the *pch1* mutation results in an early flowering phenotype under long days, while PCH1ox3 flowers later than wild type (*Figure 3C*). Unlike *elf4-2* plants, which flower early under both long days and short days (*Doyle et al., 2002*), the flowering time of *pch1* or PCH1ox3 is not different from wild type under short days (*Figure 3D*). Together, the results show that PCH1 is an output rather than a component of the circadian clock and that the *pch1* mutant is still sensitive to photoperiod in respect to flowering control.

## AP-MS using PCH1-His$_6$-FLAG$_3$ co-purified clock and light signaling components

Previous AP-MS studies with the EC components ELF3 and ELF4 identified PCH1 as a co-precipitating protein (*Huang et al., 2015*). PCH1-associated proteins were identified by AP-MS from PCH1ox3 plants expressing PCH1-His$_6$-FLAG$_3$, which are harvested at dusk (ZT12). After excluding non-specific binding proteins from negative control GFP-His$_6$-FLAG$_3$ AP-MS and non-specific structural, metabolic, and photosynthetic proteins (*Huang et al., 2015*; *Mellacheruvu et al., 2013*), PCH1 AP-MS identified 17 proteins from three biological replicate purifications (*Table 1*, for all co-purified proteins, see *Table 1—source data 1*). 15 of the 17 proteins that co-precipitated with PCH1 overlap with the ELF3 AP-MS (*Huang et al., 2015*), including the EC components ELF3, ELF4, and LUX, all five phytochromes (A to E), TANDEM ZINC KNUCKLE/PLUS3 (TZP), DAYSLEEPER, MUT9-LIKE KINASE 2 (MLK2), CHLOROPLAST RNA BINDING (CRB), the protease RD21a, and the COP1-SPA1 complex (*Table 1*). COP1-SPA1 is part of a complex that mediates the light-dependent turnover of light signaling components (*Saijo et al., 2003*). TZP positively regulates morning-specific plant growth and flowering responses through associating with phyB (*Kaiserli et al., 2015*; *Loudet et al., 2008*). DAYSLEEPER is a hAT transposase that is required for proper embryonic development (*Bundock and Hooykaas, 2005*). MLK2 is a nuclear kinase that regulates circadian rhythms and osmotic stress responses (*Huang et al., 2015*; *Wang et al., 2015*). CRB is a RNA binding protein that regulates circadian rhythms (*Hassidim et al., 2007*). RD21a is a drought-inducible cysteine protease (*Koizumi et al., 1993*). FAR-RED ELONGATED HYPOCOTYL 1 (FHY1) and TOPLESS (TPL)

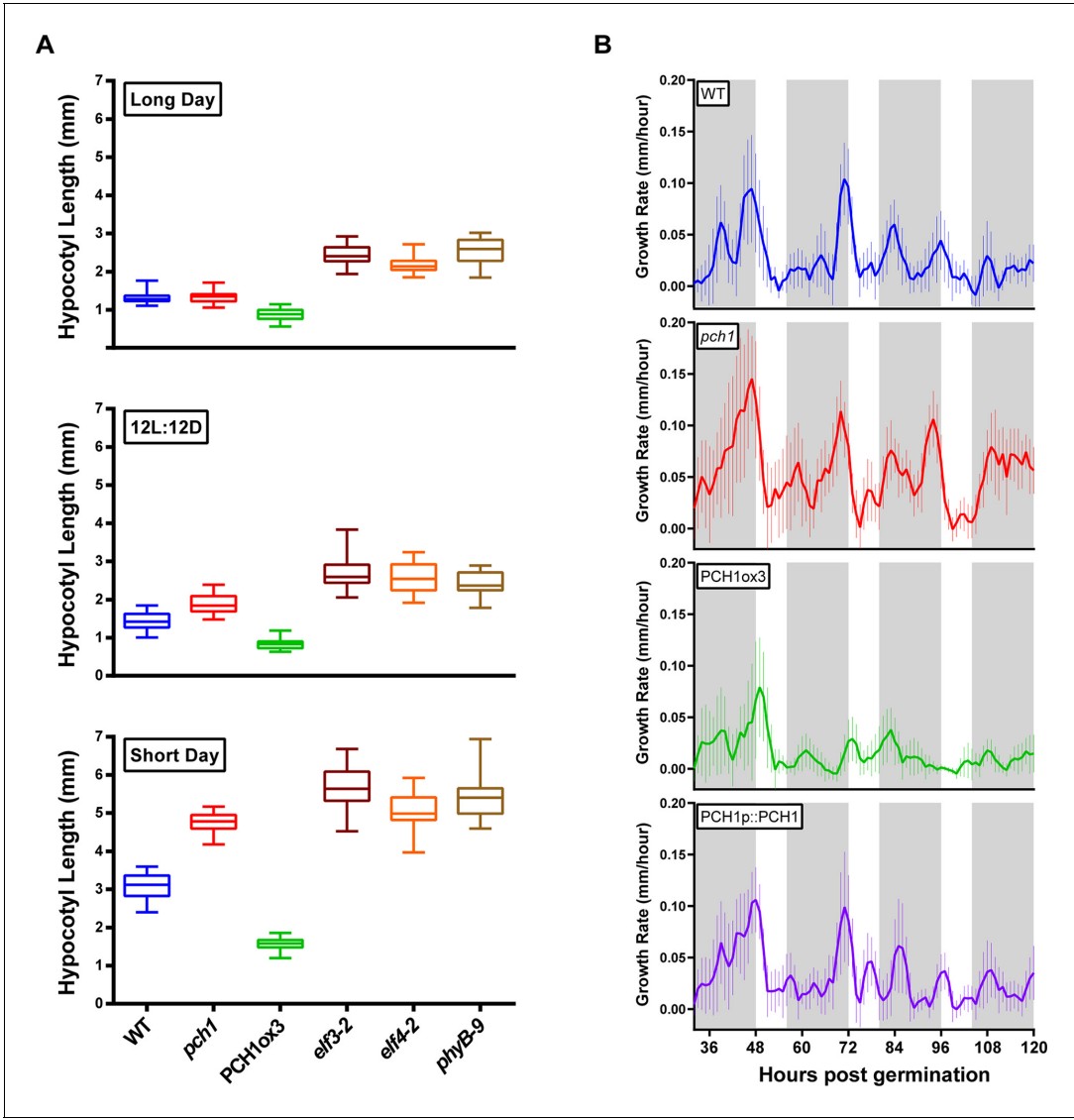

**Figure 2.** PCH1 regulates the photoperiodic response of hypocotyl elongation in the evening. (**A**) Hypocotyl lengths of 4-day-old WT, *pch1*, PCH1ox3, *elf3-2, elf4-2* and *phyB-9* seedlings grown under long day, 12L:12D and short day conditions. Mean ± 95% confidence interval (CI) (n=20). (**B**) *pch1* grows faster than WT during night. Time-lapse images were taken every hour for each seedling of WT, *pch1*, PCH1ox3 and PCH1p::PCH1 grown under short day conditions. Growth rate was calculated as the hypocotyl increase per hour and plotted against time. Solid lines are the regression analyses of data. Mean ± SEM (n ≥ 14). Grey shading indicates dark period. Also see *Figure 2—source data 1* and *2*.

The following source data and figure supplement are available for figure 2:

**Source data 1.** Raw measurements of hypocotyl lengths for *Figure 2A*.
**Source data 2.** ANOVA analyses and Bonferroni's multiple comparison tests for *Figure 2A*.
**Figure supplement 1.** *PCH1* levels regulate hypocotyl length under short day conditions.

were proteins co-purified with PCH1 that were not identified in ELF3 AP-MS. FHY1 interacts with phyA and is required for phyA nuclear import upon light treatment (*Hiltbrunner et al., 2005*). TPL is a Groucho/Tup1-type transcriptional co-repressor that interacts with proteins from circadian and development pathways (*Liu and Karmarkar, 2008*; *Wang et al., 2012*). In summary, our PCH1 AP-MS results confirm that PCH1 is a component of a reported protein-protein interaction network consisting of the EC, phytochromes and the COP1-SPA1 complex.

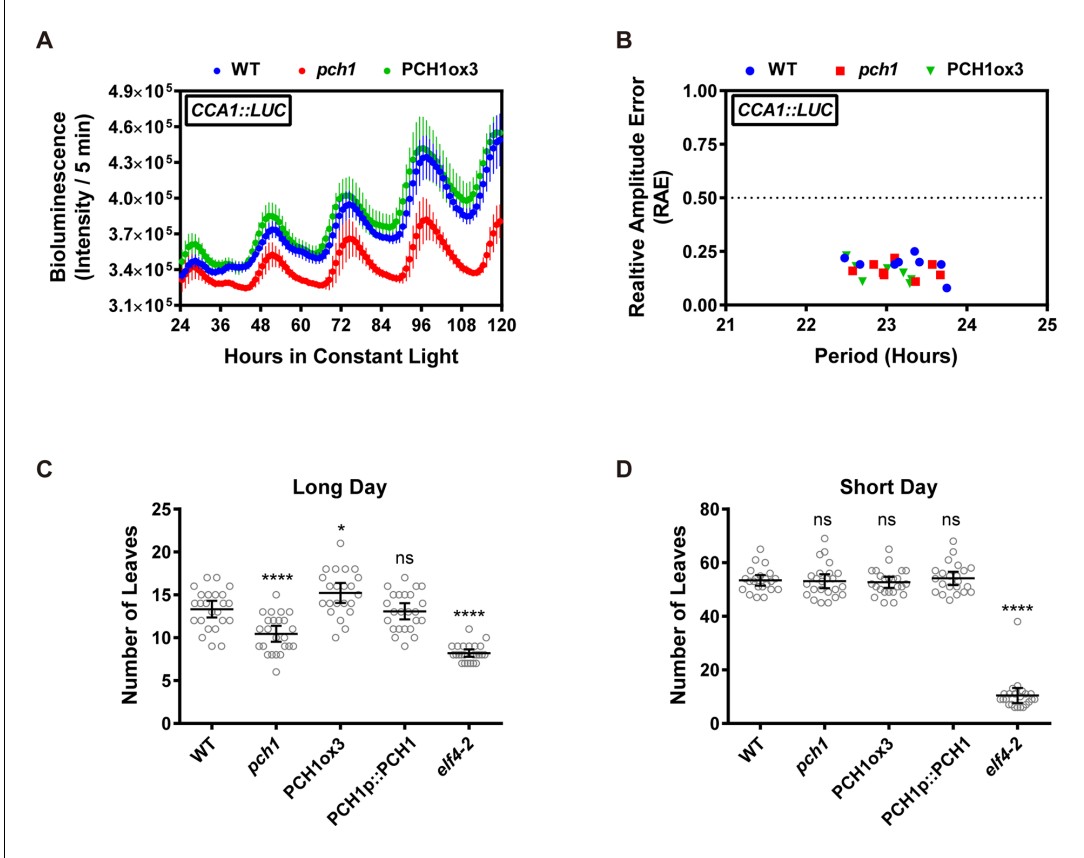

**Figure 3.** Phenotypic characterization of *pch1* and PCH1ox3 in circadian and flowering pathways. (**A**) Seedlings of WT, *pch1*, or PCH1ox3 carrying the *CCA1:LUC* luciferase reporter were grown under 12L:12D conditions for five days before transferring to continuous white light. Bioluminescence were plotted against ZT hours. Mean ± SD (n = 8). Experiments were repeated at least three times. (**B**) Relative amplitude error (RAE) versus period of WT, *pch1*, and PCH1ox3 rhythms was plotted. RAE = 0.5 was used as a cutoff (dotted line), above which a seedling is not considered rhythmic (n = 8). Experiments were repeated at least three times. (**C**) and (**D**) Flowering assays under either long day (**C**) or short day (**D**) conditions were conducted. Number of rosette leaves from WT, *pch1*, PCH1ox3 and PCH1p::PCH1 plants with 1 cm inflorescence stem was counted. Mean ± 95% CI (n $\geq$ 20). One-way ANOVA and multiple comparisons were done, with star symbols indicating if it is significantly different from WT (*p=0.012, ****p<0.0001, ns = not significantly different). Experiments were repeated twice.

The following figure supplement is available for figure 3:

**Figure supplement 1.** Modulating PCH1 levels does not affect the circadian period.

## phyB recruits PCH1 to the EC-phytochrome-COP1 interactome

ELF3, phyB, and COP1 interact with each other to form a 'triangle core' of the EC-phytochrome-COP1 interactome (*Jang et al., 2010*; *Liu et al., 2001*; *Yu et al., 2008*), recruiting other proteins into the interaction network (*Huang et al., 2015*). To determine if the association between PCH1 and other co-purified proteins depended on the EC or phyB, PCH1 AP-MS analysis in wild type (PCH1ox3) was compared to those in *elf4-2, elf3-2* or *phyB-9* backgrounds. Although PCH1 was originally found co-precipitating with ELF4 and ELF3, both are dispensable for PCH1 to associate with the light signaling components in the network (*Table 2*). In comparison, phyB is critical for recruiting PCH1 to the EC-phytochrome-COP1 interactome. In *phyB-9*, PCH1AP-MS did not co-precipitate the EC, the COP1-SPA1 complex, TZP, MLK2, RD21a, TPL, or phyC (*Table 1*). However, the association with DAYSLEEPER, CRB, phyD, phyE, phyA, and FHY1 was retained in *phyB-9*. Therefore, our PCH1 AP-MS analysis in *phyB-9* suggests that the association of PCH1 with the EC, the COP1-SPA1 complex, MLK2, and TZP is bridged by phyB, while loss of phyC could be due to a reduction in phyC caused by the *phyB* mutation (*Clack et al., 2009*). Together, our PCH1 AP-MS analyses in different

genetic backgrounds demonstrate that PCH1 is integrated into the EC-phytochrome-COP1 interactome in vivo through the association with phyB.

## PCH1 directly interacts with phyB, and preferentially binds the Pfr form

Next, yeast two-hybrid assays were used to determine if interactions between PCH1 and selected PCH1-associated proteins were direct. Consistent with the AP-MS data, direct interactions between PCH1 and ELF3, ELF4, LUX, COP1 or TZP were not observed (*Figure 4A*). However, PCH1 interacted with the C-terminus of phyB (*Figure 4A*). PCH1 also interacted with the C-terminal tail of phyD and phyE, but not with either phyA or phyC in yeast (*Figure 4B*). To validate the PCH1-phyB interaction in planta, PCH1-His$_6$-FLAG$_3$ was transiently co-expressed with a phyB-GFP fusion protein in tobacco (*Nicotiana benthamiana*) leaves. phyB-GFP specifically co-precipitated with PCH1-His$_6$-FLAG$_3$ in an anti-FLAG immunoprecipitation, while PCH1 and GFP alone did not interact (*Figure 4C*).

Light absorption by phytochromes alters their confirmation, subcellular localization, and binding to signaling partners (*Burgie et al., 2014*; *Kikis et al., 2009*; *Kircher et al., 2002*). We therefore examined the light sensitivity of the PCH1-phyB interaction. PCH1ox3 seedlings were used for phyB co-precipitation reactions after 12L:12D entrainment under white light conditions (WL). Endogenous phyB co-precipitated with PCH1 at the end of the light treatment (ZT12), confirming the AP-MS results (*Figure 4D*, lane 3). Less phyB was co-precipitated with PCH1-His$_6$-FLAG$_3$ under extended dark periods (24 or 48 hr in dark, lane 4 and 5), although the levels of phyB in the input were increased in these conditions, indicating that light promotes the PCH1-phyB interaction. The red-light sensitivity of the PCH1-phyB interaction was tested and found that 12 hr of red light treatment on the last day was sufficient to maintain the PCH1-phyB interaction (*Figure 4D*, lane 6), suggesting that PCH1 bound to the active Pfr form of phyB. Conversely, a ten-minute pulse of far-red light at the end of day (EOD-FRp) that converted phyB to the inactive Pr form reduced the PCH1-phyB interaction (*Figure 4D*, lane 7).

To test directly if PCH1 preferentially binds the active Pfr form of phyB, a reconstituted light-induced in vitro binding assay was assembled with recombinant PCH1-His$_6$-FLAG$_3$ purified from *E. coli* as bait. phyB-HA was expressed and translated in rabbit reticulocyte lysate and either the apo-protein, or the phyB holoprotein (mixed with the chromophore phytochromobilin, PΦB) were then mixed with PCH1-His$_6$-FLAG$_3$ under dark or red/far-red light, respectively. PCH1 weakly interacts with either the apoprotein or the Pr form of phyB but preferentially binds the active Pfr form of phyB, compared to the YPet (a YFP variant) control (*Figure 4E*). In summary, PCH1 directly interacts with phyB, and the interaction is light- and wavelength-sensitive in vivo and in vitro. Combined with our PCH1 AP-MS analyses in different genetic backgrounds, our protein-protein interaction/association data demonstrate that PCH1 is a new phyB-interacting protein and is integrated into the EC-phytochrome-COP1 interactome in vivo through the association with phyB (*Figure 4—figure supplement 1*).

## PCH1 localizes in the nucleus and stabilizes phyB photobodies in the early evening

A subcellular localization tool (*Kosugi et al., 2009*) identified a bipartite nuclear localization signal in PCH1 (highlighted in *Figure 1—figure supplement 5*). Transient expression of a PCH1-YPet fusion in tobacco showed that PCH1 was exclusively localized in the nucleus, while the YPet control was localized to both nucleus and cytoplasm (*Figure 5A*). Furthermore, PCH1-YPet was localized to sub-nuclear foci (*Figure 5A*) similar to the photobodies that phytochromes form after light exposure (*Kircher et al., 2002*). Indeed, when PCH1-YPet and phyB-CFP were co-expressed, they co-localized into nuclear photobodies (*Figure 5A*).

Photobodies containing phyB are necessary to suppress hypocotyl elongation in the early evening, and phyB lacking C-terminal tails can neither form photobodies nor properly regulate growth after transfer to dark conditions (*Van Buskirk et al., 2014*). Since PCH1 accumulates towards the early evening to suppress hypocotyl elongation, interacts with the C-terminus of phyB, and localizes with phyB to photobodies, we hypothesized that PCH1 regulates phyB photobody assembly/disassembly. A phyB-GFP fusion protein (PBG) was introduced into *phyB-9* and crossed into *pch1* and PCH1ox3 plants. Photobody formation was examined in short day-entrained (under 10 μmol·m$^{-2}$·s$^{-1}$

**Table 1.** Proteins Co-Purified by PCH1 AP-MS in WT and *phyB-9*. Proteins co-purified with PCH1 were identified from affinity purification coupled with mass spectrometry (AP-MS) analyses using 12L:12D grown, 10-day-old PCH1ox3 plants (in either WT or *phyB-9* mutant backgrounds) harvested at ZT12.

| | | | Exclusive unique peptide count/Percent coverage[a] | | | | |
| | | | PCH1ox3 in WT | | | PCH1ox3 in *phyB-9* | |
| AGI number | Protein name | ELF3 AP-MS[b] | rep1 | rep2 | rep3 | rep1 | rep2 |
|---|---|---|---|---|---|---|---|
| At2g16365 | PCH1[c] | Y | 30/73% | 41/85% | 37/79% | 40/85% | 32/77% |
| At2g18790 | phyB | Y | 46/69% | 47/65% | 41/60% | — | — |
| At5g35840 | phyC | Y | 31/44% | 23/28% | 25/29% | — | — |
| At4g16250 | phyD | Y | 22/47% | 19/34% | 20/38% | 22/38% | 6/11% |
| At4g18130 | phyE | Y | 41/55% | 40/52% | 45/60% | 49/60% | 31/41% |
| At1g09570 | phyA | Y | 31/46% | 36/49% | 35/45% | 36/48% | 29/39% |
| At2g37678 | FHY1 | N | 2/22% | 2/21% | 4/28% | 4/28% | 2/21% |
| At3g42170 | DAYSLEEPER | Y | 5/13% | — | 4/11% | 3/8% | 2/6% |
| At1g09340 | CRB | Y | —[d] | —[d] | 4/17% | 6/24% | 3/13% |
| At5g43630 | TZP | Y | 9/15% | 6/12% | 12/23% | — | — |
| At2g32950 | COP1 | Y | 7/15% | 8/16% | 8/18% | — | — |
| At2g46340 | SPA1 | Y | 8/14% | 5/7% | 8/12% | — | — |
| At2g25930 | ELF3 | Y | 6/12% | 11/25% | 12/26% | — | — |
| At2g40080 | ELF4 | Y | —[d] | 4/60% | 3/42% | — | — |
| At3g46640 | LUX | Y | 2/6% | —[d] | 4/15% | — | — |
| At3g03940 | MLK2 | Y | —[d] | 2/6% | 2/6% | — | — |
| At1g15750 | TPL | N | —[d] | 3/3% | 4/5% | 2/2% | — |
| At1g47128 | RD21a | Y | —[d] | 3/8% | 2/4% | —[d] | —[d] |

Also see *Table 1—source data 1*

[a] All listed proteins match 99% protein threshold, minimum number peptides of 2 and peptide threshold as 95%. Proteins not matching the criteria were marked with "—".

[b] ELF3 AP-MS (*Huang et al., 2015*) was used for comparison.

[c] Percent coverage for PCH1 is calculated using protein encoded by *At2g16365.2*.

[d] Only one exclusive unique peptide was detected.

Source data 1. The full list of proteins identified by AP-MS, listing unique peptides and the percent coverage. The full list is generated and exported by Scaffold (Proteome Software Inc., Portland, Oregon; v.4.4.3) showing all co-purified proteins from all replicates of PCH1ox3 AP-MS and the GFP Control. The file contains reports on exclusive unique peptide counts and percent coverage for each co-purified proteins, with their names, accession numbers and molecular weight.

red light) seedlings at and after the transition to dark (0, 4, 8 and 16 hr in dark or ZT56, 60, 64, and 72, respectively). In wild type (*phyB-9*/PBG), phyB-GFP formed large (>1 µm³) photobodies (PB) after 8 hr of light treatment that gradually dissembled into smaller photobodies (<1 µm³) and a diffuse nuclear GFP signal after 8 hr in dark (*Figure 5B to D*). PCH1ox3 lines showed more large phyB photobodies at the end of the day and throughout the night compared to wild type (*Figure 5B and C*). In contrast, *pch1* mutants exhibited a significant decrease in large photobodies (p<0.0001) and a significant increase in small photobodies (p<0.0001) at the dusk transition and during the first four hours of night compared to the wild type (*Figure 5C and D*). In higher red light (40 µmol·m⁻²·s⁻¹), formation of large phyB photobodies in *pch1* was significantly less (p<0.0001) at the dusk transition compared to wild type, and significantly more small photobodies was observed throughout the night (ZT56, p=0.024 and 72, p=0.0125, ZT60 and 64, p<0.0001) (*Figure 5—figure supplement 1*). However, PCH1ox3 lines showed significantly more large phyB photobodies (p≤0.0002) for all time

**Table 2.** Proteins Co-Purified by PCH1 AP-MS in *elf4-2* and *elf3-2*, compared to WT. Proteins co-purified with PCH1 were identified from affinity purification coupled with mass spectrometry (AP-MS) analyses using 12L:12D grown, 10-day-old PCH1ox3 plants in either *elf4-2* or *elf3-2* mutant backgrounds harvested at ZT12.

| | | | Exclusive unique peptide count/Percent coverage[a] | | | | | | |
| | | | PCH1ox3 in WT[c] | | | PCH1ox3 in *elf4-2* | | PCH1ox3 in *elf3-2* | |
| AGI number | Protein name | ELF3 AP-MS[b] | rep1 | rep2 | rep3 | rep1 | rep2 | rep1 | rep2 |
|---|---|---|---|---|---|---|---|---|---|
| At2g16365 | PCH1[d] | Y | 30/73% | 41/85% | 37/79% | 29/77% | 34/78% | 42/82% | 36/82% |
| At2g18790 | phyB | Y | 46/69% | 47/65% | 41/60% | 47/70% | 31/46% | 42/63% | 40/56% |
| At5g35840 | phyC | Y | 31/44% | 23/28% | 25/29% | 30/43% | 13/16% | 20/28% | 15/18% |
| At4g16250 | phyD | Y | 22/47% | 19/34% | 20/38% | 20/42% | 12/25% | 20/37% | 16/30% |
| At4g18130 | phyE | Y | 41/55% | 40/52% | 45/60% | 42/57% | 37/50% | 43/55% | 40/53% |
| At1g09570 | phyA | Y | 31/46% | 36/49% | 35/45% | 32/47% | 24/34% | 34/49% | 27/38% |
| At2g37678 | FHY1 | N | 2/22% | 2/21% | 4/28% | —[e] | 3/21% | —[e] | 3/21% |
| At3g42170 | DAYSLEEPER | Y | 5/13% | — | 4/11% | 3/7% | 5/11% | 2/5% | —[e] |
| At1g09340 | CRB | Y | —[e] | —[e] | 4/17% | 2/9% | 5/22% | 3/13% | 3/13% |
| At5g43630 | TZP | Y | 9/15% | 6/12% | 12/23% | 4/7% | —[e] | 12/21% | 2/3% |
| At2g32950 | COP1 | Y | 7/15% | 8/16% | 8/18% | 3/7% | —[e] | 12/25% | 7/11% |
| At2g46340 | SPA1 | Y | 8/14% | 5/7% | 8/12% | 5/10% | 2/4% | 17/26% | 7/11% |
| At2g25930 | ELF3 | Y | 6/12% | 11/25% | 12/26% | 4/9% | 3/6% | — | — |
| At2g40080 | ELF4 | Y | —[e] | 4/60% | 3/42% | — | — | — | —[e] |
| At3g46640 | LUX | Y | 2/6% | —[e] | 4/15% | —[e] | — | — | — |
| At3g03940 | MLK2 | Y | —[e] | 2/6% | 2/6% | — | — | 2/4% | — |
| At1g15750 | TPL | N | —[e] | 3/3% | 4/5% | 2/3% | 3/3% | 4/5% | 4/5% |
| At1g47128 | RD21a | Y | —[e] | 3/8% | 2/4% | —[e] | 2/8% | 3/8% | —[e] |

Also see *Table 1—source data 1*

[a] All listed proteins match 99% protein threshold, minimum number peptides of 2 and peptide threshold as 95%. Proteins not matching the criteria were marked with "—".

[b] ELF3 AP-MS (*Huang et al., 2015*) was used for comparison.

[c] PCH1ox3 in WT is as shown in *Table 1*, for comparison with PCH1ox3 in *elf4-2* and *elf3-2*.

[d] percent coverage for PCH1 is calculated using protein encoded by *At2g16365.2*

[e] only one exclusive unique peptide was detected.

points (*Figure 5—figure supplement 1*). These observations demonstrate that PCH1 levels regulate the fluence-dependent formation and maintenance of large phyB photobodies after illumination.

## *pch1* causes defects in red light and up-regulates expression of downstream transcription factors

The PCH1-phyB interaction prompted us to test if *pch1* results in red-light specific growth defects. The hypocotyls of wild type, *pch1*, PCH1ox3, PCH1p::PCH1, and *phyB-9* seedlings were measured under constant red light of various intensities. Compared with wild type, *pch1* seedlings have longer hypocotyls and are hyposensitive to red-light-mediated suppression of hypocotyl elongation. This phenotype was rescued in PCH1p::PCH1 transgenic plants (*Figure 6A*). Conversely, PCH1ox3 plants showed hypersensitivity to red light under all light fluences (*Figure 6A*). In either constant far-red or blue light, hypocotyl lengths of *pch1* and PCHox3 seedlings resembled those of wild type plants (*Figure 6—figure supplement 1*). These data suggest that PCH1 specifically modulates hypocotyl elongation in response to red light.

To better understand the mechanism underlying PCH1-mediated suppression of hypocotyl growth, the effect of altered *PCH1* levels on the expression of transcription factors downstream of phytochrome signaling was measured. *LONG HYPOCOTYL IN FAR RED 1 (HFR1)* and the homeobox transcription factor *ATHB-2* are two transcription factors that are regulated by phytochromes in

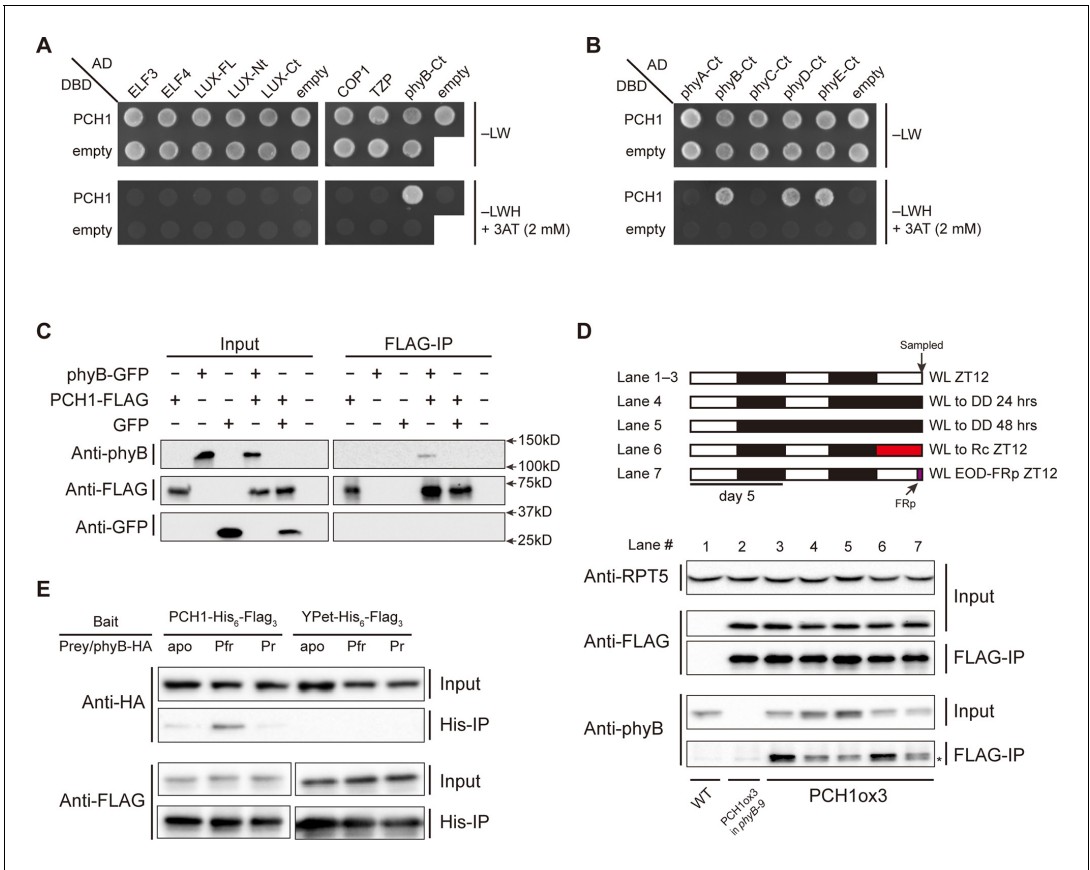

**Figure 4.** PCH1 directly interacts with phyB in a light-dependent manner. (**A**) and (**B**) yeast two-hybrid between PCH1 (fused to GAL4 DNA binding domain, DBD) and preys (ELF3, ELF4, N-/C- termini (Nt or Ct) and full length (FL) LUX, COP1, TZP and the Ct of phyA, B, C, D, and E fused to GAL4 activating domain, AD). –LW select (minus Leu and Trp) for presence of both DBD and AD constructs and–LWH+3AT plates (minus Leu, Trp and His, with 2 mM 3AT added) tested interactions. (**C**) Transient tobacco co-immunoprecipitation (IP) assay with PCH1-His$_6$-FLAG$_3$ and phyB-GFP or GFP. IPs were done against FLAG followed by westerns using either anti-FLAG, phyB or GFP antibodies. (**D**) The in-vivo PCH1-phyB interaction is light-sensitive. A schematic of the light treatment is above western. PCH1ox3 seedlings entrained in 12L:12D white light (WL) were either exposed to WL for 12 hr (WL ZT12, lane 1 to 3), subjected to extended dark (WL to DD) for 24 or 48 hr (lane 4 and 5), red light for 12 hr (WL to Rc ZT12, lane 6), or an end-of-day far-red pulse for 10 min after 12 hr of WL (WL EOD-FRp ZT12, lane 7). WT and PCH1ox3 in *phyB-9* plants are western controls. IPs were done against FLAG followed by westerns using either anti-FLAG or phyB antibodies. Anti-RPT5 was used as a loading control. The asterisk at the FLAG-IP / anti-phyB notes an unspecific band that migrates faster than phyB that is present in every lane. (**E**) PCH1 preferentially binds the Pfr form of phyB in in vitro. Recombinant His$_6$-PCH1-His$_6$-FLAG$_3$ or His$_6$-YPet-His$_6$-FLAG$_3$ was incubated with phyB-HA transcribed and translated by rabbit-reticulate lysate. PΦB absent (apo) phyB precipitations were incubated in the dark, while red (Pfr) or far red light (Pr) were incubated with 20 µM PΦB. His-affinity capture was followed by immunoblotting for anti-HA or anti-FLAG.
The following figure supplement is available for figure 4:

**Figure supplement 1.** Interaction map of PCH1-associated proteins.

response to shade and short days, and are positively correlated with hypocotyl elongation (*Kunihiro et al., 2011*; *Lorrain et al., 2008*; *Steindler et al., 1999*). qPCR analyses were done using time-course cDNA samples of short-day grown wild type, *pch1* and PCH1ox3 seedlings. In wild type, transcripts of *HFR1* and *ATHB-2* are suppressed by daylight and accumulate in the evening, with a peak at dawn (*Figure 6B*). *pch1* mutants up-regulated *HFR1* and *ATHB-2* during the dark period, in agreement with our growth rate data showing the acceleration of hypocotyl growth in *pch1* during night (*Figure 6B*). Conversely, overexpression of PCH1 suppresses *HFR1* and *ATHB-2* transcript levels throughout the light/dark cycle (*Figure 6B*). These data demonstrate that phytochrome photo-perception and downstream gene expression is regulated by PCH1.

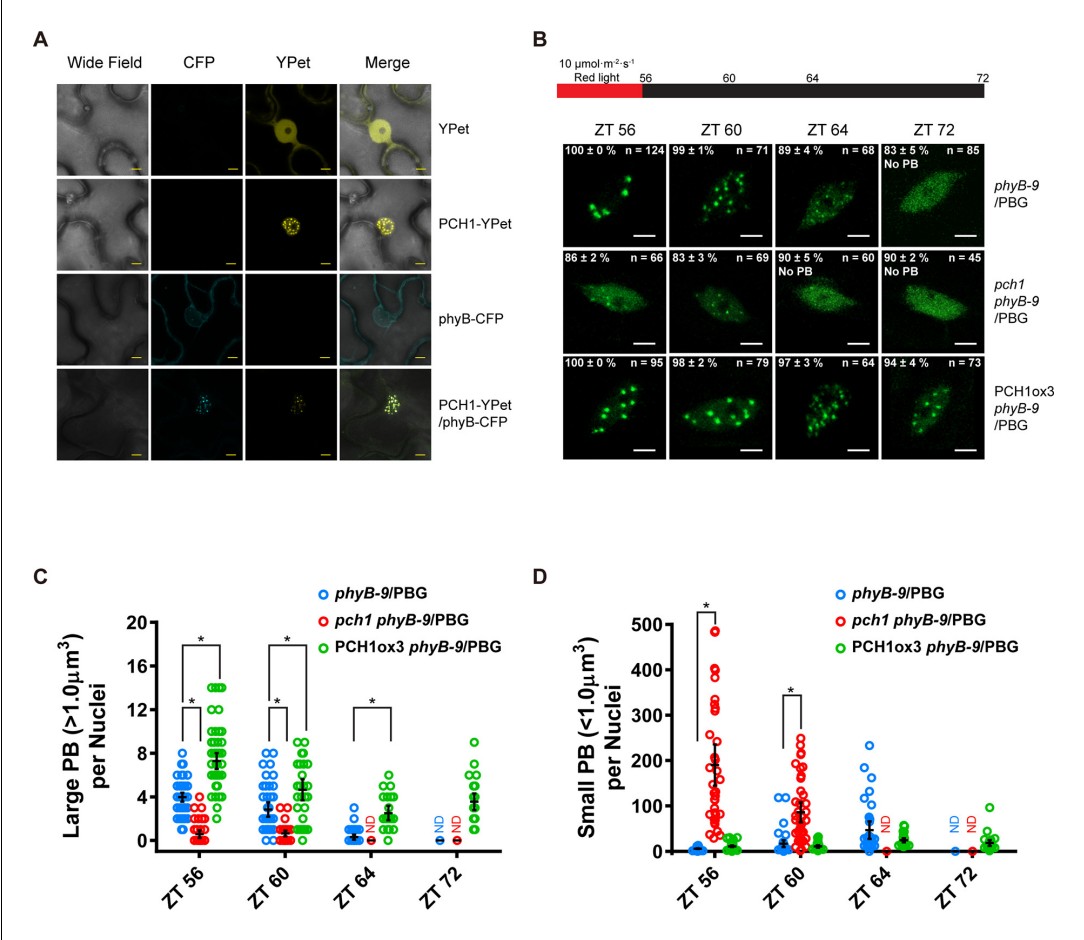

**Figure 5.** PCH1 is localized in the nucleus to stabilize phyB-containing photobodies. (**A**) PCH1-YPet is nuclear localized when transiently expressed in tobacco and co-localizes with phyB-CFP to photobodies. YPet alone was used as control. Scale bars = 25 µm. (**B**) Representative confocal images showing phyB-GFP-containing photobodies in *phyB-9*, *pch1 phyB-9* and PCH1ox3 *phyB-9* plants at indicated time points during light-to-dark transition. Plants were entrained in short days with 10 µmol·m⁻²·s⁻¹ of red light for two days before transferring to extended dark (ZT 56 to 72). The representative images were picked based on the photobody morphology of the majority of the nuclei (>50%). The percentage of nuclei showing the corresponding photobody patterns (with or without photobodies) were calculated based on three independent experiments. N represents the total number of nuclei analyzed for each time point. Scale bars equal to 5 µm. No PB = photobodies not detected. (**C**) and (**D**) compare quantitative measurements of large (>1 µm³, C) or small (<1 µm³, D) phyB photobodies in all backgrounds. Mean ± 95% CI (n ≥ 29). ND = no PB of according size were detected. * symbol indicates significantly different (p<0.05, see text for more details about each p value).

The following figure supplement is available for figure 5:

**Figure supplement 1.** Fewer large photobodies were detected in *pch1* with higher intensity of red light treatment.

### *pch1* affects PIF4 levels and PIFs are required for PCH1-mediated hypocotyl suppression

As PCH1 modulates phyB photobodies (*Figure 5*) and red light perception (*Figure 6A*), we sought to determine if altered regulation of the PIFs underlies the gene expression and growth defects observed in *pch1*. Since PIF4 directly interacts with phyB and regulates *HFR1* and *ATHB-2* expression under shade or short day conditions, *PIF4* expression and PIF4 levels were analyzed in *pch1* mutants (*Lorrain et al., 2008*; *Lorrain et al., 2009*; *Soy et al., 2012*). *PIF4* mRNA levels were upregulated in *pch1* compared to wild type in a qPCR assay (*Figure 7A*). Using a *pif4*/PIF4p::PIF4-HA line to detect PIF4 protein levels, we observed higher PIF4 levels in the evening in *pch1* compared to wild type (*Figure 7B*), suggesting that PCH1 can modulate PIF4 levels in the early evening. To elucidate if PCH1 regulates hypocotyl elongation also through other PIFs, genetic interactions between *pch1*

and *pifs* (*pif3* and *pif4 pif5*) were tested by measuring hypocotyls of seedlings grown in short days (*Figure 7C*). Single and higher order *pif* mutants reduced hypocotyl length in the wild type background, as previously reported (*Soy et al., 2012*). Introducing *pif* mutant alleles into the *pch1* mutant background progressively ameliorated the elongated hypocotyl phenotype of the *pch1* mutant. Taken together, these results show that altered PIF levels underlie the growth defects seen in *pch1*, and that *PIF3, 4,* and *5* are required for the hypocotyl growth defects in *pch1*.

## Discussion

### PCH1 regulates phyB signaling by stabilizing phyB-photobody formation

Here we show that PCH1 is a new phytochrome interacting protein that functions to increase sensitivity to red light and prolongs phyB activity by maintaining photobody formation. PCH1 binding to the C-terminus of phyB likely stabilizes the Pfr conformer (*Chen et al., 2005*), thereby providing a molecular memory of light exposure to prevent inappropriate elongation in response to long nights.

Recent models of phyB signaling and photoconversion postulated that specific binding of a yet identified factor to phyB in the Pfr state might prevent dark-reversion and maintain phyB to photobodies to sustain an active pool of phyB in the dark in vivo (*Klose et al., 2015*). We found that loss of PCH1 severely attenuated formation of large photobodies, particularly at low fluence red light (*Figure 5B*). In both low and high red-light conditions, *pch1* mutants had more small photobodies, suggesting that PCH1 regulates either the transition from small to large photobody or the maintenance of large photobodies. The phenotypes of *pch1* mutants are distinct from mutations in HEMERA, which is necessary for photobody initiation (*Chen et al., 2010*). Conversely, constitutive overexpression of PCH1 resulted in an increase in the number and prolonged maintenance of large photobodies during the night at both high and low light intensity compared to wild type (*Figure 5* and *Figure 5—figure supplement 1*). Altering *PCH1* levels, however, does not induce a constitutively photomorphogenic phenotype (*Figure 6A* and *Figure 6—figure supplement 1*). We favor a model wherein binding of PCH1 to phyB after light exposure traps phyB in an active conformation and prolongs phyB localization to large photobodies by either slowing dark reversion rates or through maintaining the superstructure of the photobody once formed.

### PCH1 prolongs red light-mediated hypocotyl suppression in the evening

Our data demonstrate that PCH1 is a new component that suppresses the photoperiodic response of hypocotyl elongation. Although PCH1 accumulates at dusk, similar to the EC, *pch1* mutants are hypersensitive to the extended night, while the EC mutants are insensitive to changing photoperiods, displaying long hypocotyls regardless of day length (*Figure 2A*). This difference is likely due to the strong transcriptional effects on *PIF4* and *PIF5* expression when the EC is absent (*Nusinow et al., 2011*). We propose that the short day-specific phenotype of *pch1* results from the coincidence of the internal clock-controlled oscillation of PCH1 and PIF4, and external photoperiodic cues. In short days, PCH1 peaks at dusk, binds photoactivated phyB and prolongs phyB photobody formation to maintain phyB in the Pfr state, which then suppresses PIF4 levels in the early evening, reducing PIF4 activities and hypocotyl growth (*Figure 8*). As the daytime increases in long days, the peak of PIF4 (at ZT8) is no longer at dusk but in the middle of the day (*Yamashino et al., 2014* and *Figure 8—figure supplement 1*), when light perception by the phytochromes would act as the major suppressor of elongation through post-translation regulation of PIF protein levels, masking the requirement of PCH1 (*Figure 8*). PCH1 overexpression lines would constantly suppress hypocotyl elongation due to maintaining phyB photobodies throughout the night (*Figure 5B*), leading to shortened hypocotyls as observed (*Figure 2A*). It is likely other PIFs (e.g. PIF3 and PIF5) also contribute to the hypocotyl phenotype of *pch1*, as suggested by genetic analysis showing that higher order *pif* mutations progressively suppresses the long hypocotyl phenotype of *pch1* (*Figure 7C*).

In summary, we have identified a new factor that binds to active phyB to extend its activity in the dark, and can maintain photomorphogenesis programs even in the long nights of short days. We anticipate that modulating PCH1 levels or its expression pattern could potentially alter light perception and lead to improved growth responses at latitudes where photoperiod changes during the

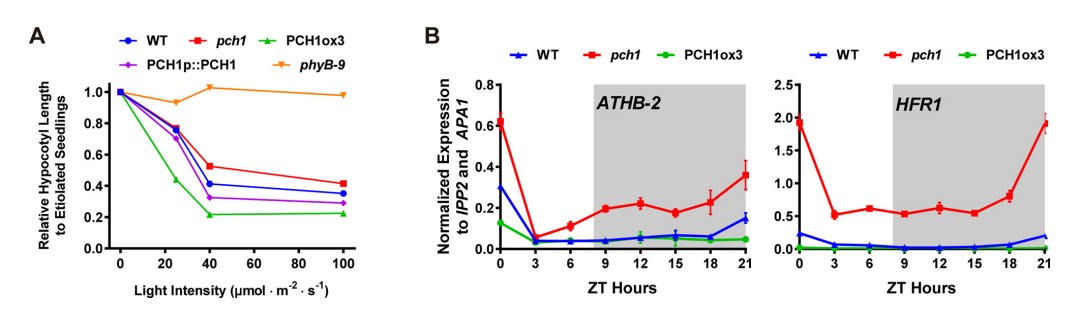

**Figure 6.** *pch1* exhibits defects in red light responsive hypocotyl growth and expression of downstream transcription factors. (**A**) Hypocotyl lengths of 4-day-old WT, *pch1*, PCH1ox3, PCH1p::PCH1 and *phyB-9* seedlings grown under either dark or constant red light of various intensities (25, 40 and 100 μmol·m$^{-2}$·s$^{-1}$). Mean ± 95% CI (n = 20). Hypocotyl lengths of light-grown seedlings were normalized to dark-grown (etiolated) seedlings, and were plotted against light intensities to generate the responsive curve. Etiolated hypocotyl lengths (mean ± SD) of WT, *pch1*, PCH1ox3, PCH1p::PCH1 and *phyB-9* are 9.02 ± 0.90, 8.47 ± 0.66, 8.04 ± 0.70, 8.71 ± 0.71 and 7.63 ± 0.79, respectively. (**B**) qPCR of *HFR1*, *ATHB-2* using time-course cDNA samples of short-day grown, 4-day-old WT, *pch1* and PCH1ox3 seedlings. Expression was normalized to *IPP2* and *APA1*. Mean ± SD (n=3 biological reps). Grey shading indicates dark period.

The following figure supplement is available for figure 6:

**Figure supplement 1.** *pch1* does not affect far-red or blue light mediated hypocotyl elongation.

agricultural season or in species whose yield is highly sensitive to photoperiod (*Sadras and Slafer, 2012*).

## Materials and methods

### Plant materials and growth conditions

All plants used in this study are in the Columbia (Col) ecotype of *A. thaliana* unless noted. *pch1* (SALK_024229), *pif3* (SALK_081927C), *phyB-9* and *phyA-211* were obtained from the ABRC and described previously (*Alonso et al., 2003*; *Reed et al., 1993*; *Ruckle et al., 2012*; *Sung et al., 2007*; *Zhong et al., 2012*). *elf3-2*, *elf4-2*, and *pif4 pif5* (*pif4-101 pif5-1*) lines were described previously (*Nusinow et al., 2011*). WS and *cry1 cry2* seeds were kindly provided by Takato Imaizumi (University of Washington, Seattle) and are in the Wassilewskija (WS) ecotype. *pif4*/PIF4p::PIF4-HA transgenic plants were kindly provided by Christian Fankhauser (University of Lausanne Center for Integrative Genomics, Switzerland) and crossed with *pch1*. Homozygous mutant plants were validated by testing luciferase bioluminescence, drug resistance, and by PCR or dCAPS-based genotyping.

Seeds were surface sterilized and plated on 1/2X Murashige and Skoog medium supplemented with 0.8% agar and 3% sucrose (w/v). Sterilized seeds on plates were then stratified for 2 to 4 days in darkness at 4°C. After stratification, plates were placed horizontally in chambers for 4 days, supplied with white light (WL, 80 μmol·m$^{-2}$·s$^{-1}$) and set to 22°C, under various photoperiodic conditions, including long day, 12L:12D and short day (Light: Dark= 16: 8, 12: 12 and 8: 16 hr, respectively). For measuring hypocotyl lengths of *pch1 and pifs* mutants, 7-day-old seedlings grown under short day conditions were compared. For monochromatic wavelength treatments, stratified seedlings were first exposed to white light (80 μmol·m$^{-2}$·s$^{-1}$) for 5 hr to synchronize germination and then were grown under constant red light (Rc, 40 μmol·m$^{-2}$·s$^{-1}$), far-red light (24 μmol·m$^{-2}$·s$^{-1}$), blue light (20 μmol·m$^{-2}$·s$^{-1}$) conditions (CLF Plant Climatics, Wertingen, Germany) or in the dark for 4 days, before hypocotyl measurements were taken.

### Phenotypic characterization and statistical analysis

For hypocotyl elongation assays, 4 to 7-day-old seedlings (as specified in each figure legend) grown under different photoperiod or light conditions were arrayed, photographed with a ruler for measuring hypocotyl length using the Image J software (NIH, Bethesda, Maryland). For measuring growth

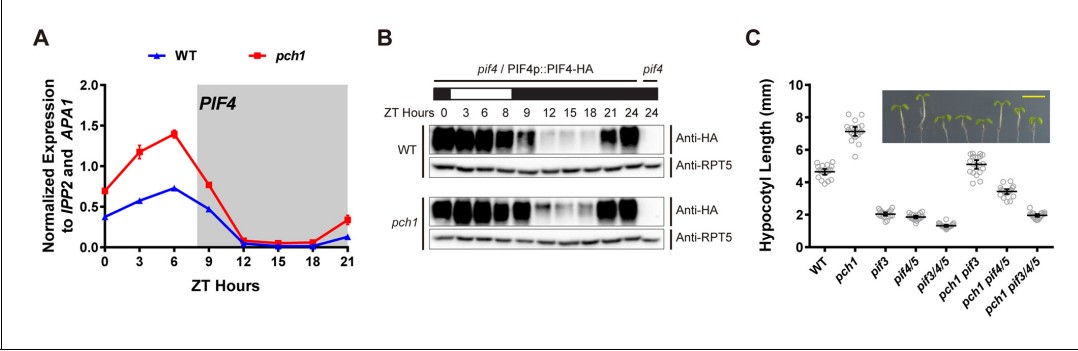

**Figure 7.** *pch1* affects PIF4 levels and PIFs are required for the hypocotyl phenotype in *pch1*. (**A**) qPCR of *PIF4* using time-course cDNA samples of short-day grown, 4-day-old WT and *pch1* seedlings. Expression was normalized to *IPP2* and *APA1*. Mean ± SD (n=3 biological reps). Grey shading indicates dark period. (**B**) Anti-HA immunoblots for testing PIF4-HA levels in WT (*pif4*/PIF4p::PIF4-HA) and *pch1* (*pch1 pif4*/PIF4p::PIF4-HA) genetic backgrounds. Time-course protein extracts (from ZT 0 to 24, with 3 hr intervals, plus ZT 8) were made from short-day-grown, 4-day-old seedlings. Rectangles above blots represent light/dark conditions under which samples were flash frozen in liquid $N_2$, white = light and black = dark. Anti-RPT5 used as a loading control. *pif4* extracts were used as a negative western control. (**C**) Hypocotyl lengths of 7-day-old, short-day-grown WT, *pch1*, *pif3*, *pif4/5*, *pif3/4/5*, *pch1 pif3*, *pch1 pif4/5* and *pch1 pif3/4/5* seedlings were measured. Mean ± 95% CI (n = 20). Inset shows representative phenotypes with the scale bar = 5 mm.

rate, a total of 96 time-lapse images were taken every hour for each seedling (grown under short day conditions) using an infrared-sensitive camera (Pi-NoIR, Amazon.com, Seattle, Washington) with a visible light cut-out filter (87, Lee Filters, Burbank, CA) and hand assembled 880 nm LED array controlled by a custom script running on a Raspberry Pi (Amazon.com, Seattle, Washington) from ZT30 to ZT125. Hypocotyl lengths were then measured by Image J (NIH, Bethesda, Maryland) to calculate growth rates using PRISM software (version 6.0, Graphpad.com, La Jolla, California). For flowering assay, number of rosette leaves from plants with 1cm inflorescence stem was counted. For characterizing clock phenotype, a luciferase-based assay using the *CCA1::LUC* reporter was monitored as described previously (*Huang et al., 2015*). Statistical analyses (one-way or two-way ANOVA analysis with Bonferroni's multiple comparisons test) for all experiments were performed using PRISM software (Graphpad, La Jolla, California, version 6.0, Graphpad.com).

## Vectors construction

pB7HFC vector was used for constitutively expressing C-terminal $His_6$-$FLAG_3$ fusion proteins (*Huang et al., 2015*). To generate the pB7SHHc and pB7YSHHc vectors (for generating PCH1-YPet fusion protein used in a transient expression assay), we first modified the pB7WG2 vector by introducing an AvrII restriction site. The pB7WG2 vector (*Karimi et al., 2002*) was used as the template for amplifying two pieces of overlapping DNA fragments with an AvrII site added. These two fragments of AvrIIA (using primers pDAN0193 and pDAN0202) and AvrIIB (using primers pDAN201 and pDAN0223) were diluted, mixed to serve as template and were re-amplified with pDAN0193 and pDAN0223 to generate a longer fragment AvrIIC with the AvrII site in the middle. The pB7WG2 plasmid was then linearized by digestion with EcoRI and XbaI and recombined with AvrIIC fragment using In-Fusion HD cloning (Clontech, Mountain View, California) to generate the pB7AVRII vector, which was verified by sequencing and served as the backbone of pB7SHHc and pB7YSHHc vectors.

DNA synthesis (gBlocks Gene Fragments, IDT, Coralville, Iowa) was used to generate a template sequence of 2xStrepII-HA-$His_6$-TEV-$FLAG_3$-TEV-$His_6$-HA-2xStrepII, which contains 2xStrepII, HA, $His_6$, Tobacco Etch Virus protease cleavage sites, and $FLAG_3$ epitopes for making all combinations of tags we need to put into the pB7AVRII vector. The sequence of this template is as follows: 5'-GGAAGCTGGAGCCACCCTCAATTTGAAAAGGGAGGAGGATCTGGAGGTGGTTCTGGTGGTGG TTCTTGGTCTCACCCACAATTCGAAAAGGGGTTCTTACCCATACGATGTTCCAGATTACGCTCA TCACCATCACCATCACGATATTCCAACTACTGCTAGCGAGAATTTGTATTTTCAGGGTGAGC TCGACTACAAAGACCATGACGGTGATTATAAAGATCATGACATCGACTACAAGGATGACGATGA- CAAGGATATACCTACTACTGCTTCTGAAAATCTGTACTTTCAGGGAGAACTGCACCATCATCATCA TCACTACCCTTACGATGTGCCAGACTACGCTGGATCTTGGTCTCATCCACAG

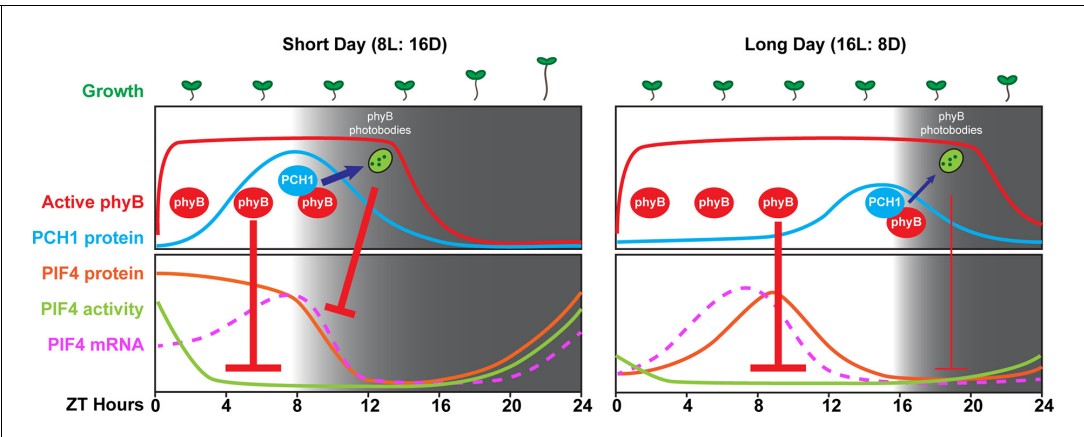

**Figure 8.** A model of PCH1-regulated day-length specific growth. A proposed model illustrates the role of PCH1 in controlling the photoperiodic hypocotyl elongation response. In short days, PCH1 peaks at dusk (ZT 8), maintains phyB photobody formation to suppress PIF4 levels and activities (downstream gene expression) to suppress hypocotyl elongation in the early evening. In long days, PIF4 peaks in the middle of the day and is repressed by active phyB. PIF4 protein decreases to basal level prior to dusk (*Figure 8—figure supplement 1* and *Yamashino et al., 2014*), therefore no longer requiring the additional suppression mediated by PCH1-regulated phyB photobodies in the evening.

The following figure supplement is available for figure 8:

**Figure supplement 1.** Expression of *PIF3, PIF4* and *PIF5* under multiple photoperiods.

TTTGAAAAGGGAGGAGGATCTGGAGGAGGATCTGGAGGAGGATCTTGGAGTCATCCTCAG TTCGAGAAG–3'. Primer set of pDAN0242 and pDAN0241 was used to amplify 2xSrepII-HA-His$_6$. The tandem tag was then recombined with pB7AVRII, which was linearized by AvrII digestion, to generate pB7SHHc using In-Fusion HD cloning (Clontech, Mountain View, California).

To generate the pB7YSHHc vector, YPet sequence was amplified from pBJ36 containing a YPet-3xHA tag (pBJ36-YPet-3xHA) as reported previously (*Krogan et al., 2012*) (a generous gift from Dr. Jeff A. Long) using primers pDAN0249 and pDAN0250 and recombined with pB7SHHc digested with AvrII using In-Fusion HD cloning (Clontech, Mountain View, California) to generate pB7YSHHc. See *Table 3* for primer sequences.

## Plasmid constructs and generation of transgenic plants

The pB7HFC vector was described previously (*Huang et al., 2015*). All cDNAs encoding either full-length or fragments of tested genes (with or without stop codons, as listed in *Table 3*) were first cloned into the pENTR/D-TOPO vector (Thermo Scientific, Waltham, Massachusetts) and were verified by sequencing. Transgenes were introduced into various genetic backgrounds by crossing.

To generate PCH1 overexpression lines (Col [*35S::PCH1-His$_6$-FLAG$_3$*][*CCA1::LUC*]), cDNA of *PCH1* (without the stop codon) was Gateway cloned (LR reaction, Invitrogen) into the pB7HFC vector. The pB7HFC-PCH1 construct was then transformed into Col [*CCA1::LUC*] plants by the floral dip method (*Clough and Bent, 1998*). Two homozygous lines PCH1ox3 and 4 were identified and used in this paper. *elf4-2, elf3-2* and *phyB-9* (all carrying the *CCA1::LUC* reporter) were crossed with PCH1ox3.

To generate the *pch1* [*PCH1pro::PCH1-His$_6$-FLAG$_3$*][*CCA1::LUC*] complementation line (PCH1p:: PCH1-7 and -8), a fragment from ~1.5 kb sequence upstream of the transcription start site plus 5'UTR to exon 1 of *PCH1* was cloned, using primers that introduced a HindIII restriction enzyme cutting site to its 5' end (listed in *Table 3*). The amplified fragment was then swapped into the pB7HFC-PCH1 construct to replace the 35S promoter by restriction enzyme digestion with HindIII and XhoI and ligation. The pB7HFC-PCH1p::PCH1 construct was then transformed into *pch1* [*CCA1::LUC*] plants. PCH1p::PCH1-7 plants was used in time-course western blottings as well as physiological assays.

cDNA of *PCH1* without the stop codon was gateway cloned into the pB7YSHHc vector to make the pB7YSHHc-PCH1 construct (*35S::PCH1-YPet-2xStrepII-HA-His$_6$*). Coding sequence of YPet was

gateway cloned into the pB7SHHc vector to serve as a control (*35S::YPet-2xStrepII-HA-His$_6$*, pB7SHHc-YPet). The GFP construct has been described previously (*35S::GFP*, pB7GFP) (*Huang et al., 2015*). The phyB-GFP construct is a generous gift from Dr. A. Nagatani (Kyoto University, Japan) that was described previously (*Yamaguchi et al., 1999*) and was transformed into *phyB-9* plants to generate *phyB-9 [35S::phyB-GFP]* plants (PBG). PBG plants were then crossed with *pch1 phyB-9* and PCH1ox3 *phyB-9* to make *pch1/*PBG and PCH1ox3/PBG lines (without the *CCA1:LUC* reporter).

The *pif4*/PIF4p::PIF4-HA transgenic line was generated by Séverine Lorrain in Christian Fankhauser's lab (University of Lausanne Center for Integrative Genomics, Switzerland), which expresses a C-terminal PIF4-3xHA fusion protein driven by the *PIF4* native promoter (~2.1 kb upstream of the start codon).

## qPCR and semi-quantitative qPCR

Time course RNA samples (with 3 hr interval) were made from 4-day-old seedlings of Col, *pch1*, and PCH1ox3 (all carrying the *CCA1::LUC* reporter) grown under short day conditions, using the RNeasy Plant Mini Kit (Qiagen, Hilden, Germany). 1 µg of total RNA was reverse transcribed to make cDNA using the iScript cDNA synthesis kit (Bio-Rad, Carlsbad, CA), which was quantified by quantitative real-time PCR (qPCR) using a CFX 384 Real-Time System (C1000 Touch Thermal Cycler, Bio-Rad, Hercules, California). PCR was set up as follows: 3 min at 95°C, followed by 40 cycles of 10 s at 95°C, 10 s at 55°C and 20 s at 72°C. A melting curve analysis was conducted right after all PCR cycles are done. Both *IPP2* (*At3g02780*) and *APA1* (*At1g11910*), expression of which remain stable during the diurnal cycle, were used as the normalization controls (*Hazen et al., 2005*; *Michael et al., 2008a*; *Nusinow et al., 2011*). PCR efficiencies for each target/reference genes were assessed and qPCR analyses were carried out by applying actual PCR efficiencies to calculate the relative expression of each sample, as described previously (*Hellemans et al., 2007*; *Remans et al., 2014*). All qPCR were done using 3 biological replicates.

For semi-quantitative qPCR, all cDNA samples of Col [*CCA1::LUC*] or *pch1 [CCA1::LUC*] time course (from ZT0 to ZT24, with 3 hr intervals) were pooled and 200 ng of pooled cDNA was used. 30 ng of genomic DNA was used as comparison. PCR conditions are as follows: 5 min at 95°C, followed by 30 cycles of 30 s at 95°C, 30 s at 55°C and 20 s at 72°C for cDNA template or 30 s at 72°C for genomic DNA template). See *Table 3* for primer sequences.

## Yeast two-hybrid analysis

We used the Matchmaker GAL4 Two-Hybrid systems (Clontech, Mountain View, California) to analyze protein-protein interactions in yeast. Verified cDNA sequences (primers listed below) were cloned into either the pAS2-GW or pACT2-GW vector, which are derived from the pAS2-1 and pACT2 plasmids of Clontech (*Nusinow et al., 2011*), through Gateway LR recombination reactions (Thermo Scientific, Waltham, Massachusetts). Both the DNA binding domain (DBD) or activating domain (AD)-fused constructs were transformed into *Saccharomyces cerevisiae* strain Y187 (*MATα*) and the AH109 (*MATa*), respectively, by the Li-Ac transformation protocol according to the yeast handbook (Clontech, Mountain View, California). Two yeast strains of the same optical density (OD$_{600}$) were mixed and incubated in low pH YCM media (1% yeast extract, 1% bactopeptone, 2% dextrose, pH 4.5) for 4.5 hr at 30°C. Afterwards, cells were transferred to regular YPDA media and incubated overnight at 30°C. Diploid yeast were then grown on CSM –Leu –Trp plates (Sunrise Science, San Diego, California) supplemented with extra Adenine (30 mg/L final concentration) for selection of bait and prey vectors and were tested for protein-protein interaction by plate replicating on CSM –Leu –Trp –His media supplemented with extra Adenine and 2 mM 3-Amino-1,2,4-triazole (3AT). Pictures were taken after 4-day incubation at 30°C. Empty pAS2-GW and pACT2-GW plasmids were used as negative controls. See *Table 3* for primer sequences.

## *N. benthamiana* transient expression

Overnight saturated cultures of *Agrobacterium tumefaciens* strain GV3101 carrying pB7YSHHc-PCH1, pB7SHHc-YPet, phyB-CFP (*35S::phyB-CFP*) (*Nito et al., 2013*), pB7HFC-PCH1, phyB-GFP (*35S::phyB-GFP*, PBG) and GFP (*35S::GFP*, pB7GFP) were diluted in 10 mM MgCl$_2$ (OD600 = 0.8) and kept at room temperature for 1~2 hr. An Agrobacterium culture of *35S:P19-HA* was also diluted

**Table 3.** Primers used in this study.

**Primers used for cloning PCH1 and PCH1 promoter** [a]

| Amplified Fragments | Forward primer (5'->3') | Reverse primer (5'->3') |
| --- | --- | --- |
| PCH1-stop | CACCATGTCTGAACATGTTATGGTTTTGG | CTACCTCAAATCCCTTGCATTCCA |
| PCH1-nonstop | CACCATGTCTGAACATGTTATGGTTTTGG | CCTCAAATCCCTTGCATTCCAAAC |
| PCH1-promoter [b] | AAGCTTAGTTTCCTCATCATTTGCTATTG | GCGTAAATCCTCACCGGTCTT |

**Primers used to generate yeast two-hybrid constructs, all with a stop codon** [a]

| Amplified fragments | Forward primer (5'->3') | Reverse primer (5'->3') |
| --- | --- | --- |
| PCH1 | CACCATGTCTGAACATGTTATGGTTTTGG | CTACCTCAAATCCCTTGCATTCCA |
| ELF3 | CACCATGAAGAGAGGGAAAGATGAG | CTAAGGCTTAGAGGAGTCATAGCGTTT |
| ELF4 | CACCATGAAGAGGAACGGCGAGACGA | TTAAGCTCTAGTTCCGGCAGCACC |
| LUX (full length) | CACCATGGGAGAGGAAGTACAAA | TTAATTCTCATTTGCGCTTCCACCT |
| LUX-Nt (amino acids 1-143) | CACCATGGGAGAGGAAGTACAAA | CTATTTAAGTGTTTTCCCAGATAG |
| LUX-Ct (amino acids 144-324) | CACCATGCGACCGCGTTTAGTGTGGACA | TTAATTCTCATTTGCGCTTCCACCT |
| phyA-Ct (amino acids 606-1123) | CACCATGGATCTCAAAATTGATGGTATACAA | CTACTTGTTTGCTGCAGCGAGTTC |
| phyB-Ct (amino acids 640-1173) | CACCATGGCGGGGGAACAGGGGATTGATGAG | CTAATATGGCATCATCAGCATCATGTCA |
| phyC-Ct (amino acids 592-1112) | CACCATGGATAATAGGGTTCAGAAGGTAGAT | TCAAATCAAGGGAAATTCTGTGAGGATCAC |
| phyD-Ct (amino acids 645-1165) | CACCATGGTACAGCAAGGGATGCAG | TCATGAAGAGGGCATCATCATCA |
| phyE-Ct (amino acids 583-1113) | CACCATGAATGGCGTAGCAAGAGATGC | CTACTTTATGCTTGAACTACCCTCTGT |
| COP1 | CACCATGGAAGAGATTTCGACGGA | TCACGCAGCGAGTACCAGAACTTTG |
| TZP | CACCATGGGAGATGGAGATGAGCAA | CTAAAAGCCTAACATTTTTCTCTGCTGA |

**Primers used for qPCR**

| Gene | Forward primer (5'->3') | Reverse primer (5'->3') |
| --- | --- | --- |
| PCH1 set A | CCGGCTCCATTTCTTCGTCA | TCCGGAACAAGAGGTGGTTCT |
| PCH1 set B | GAAGTTATTGTTGTCGCCCT | GGGAAATCCAAAGCGGTATT |
| IPP2 | CTCCCTTGGGACGTATGCTG | TTGAACCTTCACGTCTCGCA |
| APA1 (At1g11910) [c] | CTCCAGAAGAGTATGTTCTGAAAG | TCCCAAGATCCAGAGAGGTC |
| HFR1 | TAAATTGGCCATTACCACCGTTTA | ACCGTGAAGAGACTGAGGAGAAGA |
| ATHB-2 | GAAGCAGAAGCAAGCATTGG | CGACGGTTCTCTTCCGTTAG |
| PIF4 | GTTGTTGACTTTGCTGTCCCGC | CCAGATCATCTCCGACCGGTTT |

**Primers for genotyping**

| Mutant | for wild type PCR (5'->3') | for mutant PCR (5'->3') |
| --- | --- | --- |
| pch1 (SALK_024229) | TGTCAGGTATTTCGGTCCTTG (LP) and CACTTGCTTGATGCTCATGAG (RP) | AAGAACCGGCAAAGATACCAC (RP) and ATTTTGCCGATTTCGGAAC (LBb 1.3) |
| pif3 (SALK_081927C) | AGTCTGTTGCTTCTGCTACGC (LP) and AAGAACCGGCAAAGATACCAC (RP) | ACATACAGATCTTTACGGTGG (RP)and ATTTTGCCGATTTCGGAAC (LBb 1.3) |
| pif4 (pif4-101) [d] | CTCGATTTCCGGTTATGG (SL42) and CAGACGGTTGATCATCTG (SL43) | GCATCTGAATTTCATAACCAATC (PD14) and CAGACGGTTGATCATCTG (SL43) |
| pif5 (pif5-1) [d] | TCGCTCACTCGCTTACTTAC (SL46) and TCTCTACGAGCTTGGCTTTG (SL47) | TCGCTCACTCGCTTACTTAC (SL46) and GGCAATCAGCTGTTGCCCGTCTCACTGGTG (JMLB1) |

*Table 3 continued*

**Primers used for cloning PCH1 and PCH1 promoter** [a]

| elf3-2 [c] | TGAGTATTTGTTTCTTCTCGAGC and CATATGGAGGGAAGTAGCCATTAC | TGGTTATTTATTCTCCGCTCTTTC and TTGTTCCATTAGCTGTTCAACCTA |
|---|---|---|
| elf4-2 [c] | ATGGGTTTGCTCCCACGGATTA and CAGGTTCCGGGAACCAAATTCT, cut with HpyCH4V. WT has 5 cuts while *elf4-2* has 4 cuts to give a unique 689 bp band. | |
| phyB-9 | GTGGAAGAAGCTCGACCAGGCTTTG and GTGTCTGCGTTCTCAAAACG, cut with MnlI, *phyB-9* gives 167+18 bp bands, WT gives a 185 bp band. | |

**Primers for making pB7SHHc and pB7YSHHc**

| Primer Name | Sequence (5'->3') |
|---|---|
| pDAN0193 | TGCCCGCCTGATGAATGCTC |
| pDAN0202 | GCGGGATATCACCACCCTAGGCACCACTTTGTACAAGAAAGCTGA |
| pDAN201 | TCAGCTTTCTTGTACAAAGTGGTGCCTAGGGTGGTGATATCCCGC |
| pDAN0223 | ATTCTCATGTATGATAATTCGAGG |
| pDAN0242 | TACAAAGTGGTGCCTAGGGGTGGAAGCTGGAGCCACCCTC |
| pDAN0241 | GCGGGATATCACCACCCTAGTGATGGTGATGGTGATGAGCG |
| pDAN0249 | GCTTTCTTGTACAAAGTGGTGCCTGCTGCTGCTGCC |
| pDAN0250 | GGTGGCTCCAGCTTCCACCCCCCCTTATAGAGCTCGTTC |

[a] CACC (underscored) were added to forward primers for cloning into the pENTR/D-TOPO vector.

[b] a Hind III restriction site (in bold) was added to the forward primer.

[c] (**Nusinow et al., 2011**).

[d] (**de Lucas et al., 2008**).

into the same concentration and mixed (at a ratio of 1:1) with each culture to suppress gene silencing (**Chapman, 2004**). The cultures were then spot-infiltrated into 4 to 5-week-old *Nicotiana benthamiana* from the abaxial side of leaves. After 48 hr, infected leaves were flash frozen for protein extraction and co-IP experiments or were cut into small square pieces, mounted in water and used for confocal microscopy.

## Confocal microscopy and quantitative measurement of phyB photobodies

For PCH1-YPet and phyB-CFP co-localization assay, confocal microscopy was performed with a Leica TCS SP8 confocal laser scanning microscope and an HC PL APO CS2 63x/1.20 WATER objective lens (Leica Microsystems, Mannheim, Germany). Light source is provided by the UV Diode laser (for CFP) or the White Light Laser (WLL, for YPet), while all emission fluorescence signals were detected by the HyD detector. CFP fluorescence was monitored by a 460–505 nm band emission and a 405 nm excitation line of the UV Diode laser, with 2% transmission value. YPet fluorescence was sequentially monitored by a 525–600 nm band emission and a 514 nm excitation line of an Ar laser, with 5% transmission value. Line average was set as 16 to reduce noise and frame accumulation was set as 1.

For measuring phyB photobodies in *phyB-9*, *pch1 phyB-9* and PCH1ox3 *phyB-9* plants expressing phyB-GFP (PBG), seedlings were sampled at ZT 56 (under light), 60 (dark), 64 (dark), and 72 (dark) for short-day-entrained (by 10 or 40 $\mu mol \cdot m^{-2} \cdot s^{-1}$ red light) seedlings. Fixation was carried out as follow steps: seedlings were first immersed in 2% paraformaldehyde in 1x PBS on ice with 15 min vacuum followed by incubation in 50 mM $NH_4Cl$ in 1xPBS for 5 min 3 times, and washed by 1xPBS with 0.2% TritonX-100 for 5 min one time and 1xPBS for 5 min 2 times. Fixed seedlings were mounted on Superfrost slides using 1x PBS. Nuclei from hypocotyl were imaged using a Zeiss LSM 510 inverted confocal microscope. GFP signal was detected using a $100\times$ Plan-Apochromat oil-immersion objective, 488-nm excitation from argon laser and 505 to 550 nm bandpass detector setting. The proportion of nuclei with or without photobodies was manually scored. To quantify the number and size of photobodies, confocal images were analyzed by Huygens Essential software. The object analyzer

tool was used to threshold the image and to calculate the volume of each photobody in the image. Total number of large photobodies (>1.0 μm$^3$) or small photobodies (< 1.0 μm$^3$) was presented.

## Protein extraction, immunoprecipitation and western blot analysis

For time-course sampling, seedlings were grown on sterilized qualitative filter paper (Whatman, Maidstone, United Kingdom) for 4 days, at 22°C under various photoperiods (long day, 12L:12D and short day). 0.5 g of PCH1p::PCH1-7 or PCH1ox3 whole seedlings was collected every 3 hr from ZT0 to ZT24 and flash frozen in liquid N$_2$. For PIF4p::PIF4-HA transgenic plants in *pif4* (WT) and in *pch1pif4*, 4-day-old seedlings grown under short day conditions at 22°C were samples from ZT0 to ZT 24, with 3 hr interval and with addition of ZT8. Each time-course sample was put in a 2 mL tube that contained three 3.2-mm stainless steel beads (Biospec Bartlesville, Oklahoma). It is noted that samples undergoing dark to light transitions (e.g. ZT0 and ZT24) were collected in the dark before the transition to light, while ZT8 samples were harvested in light. For co-IP experiments testing phyB-PCH1 interaction under different light treatments (light, dark, red light and end-of-day far red light treatments), seedlings were grown under 12L:12D conditions at 22°C, on sterilized qualitative filter paper (Whatman, Maidstone, United Kingdom) for four days and sampled at specific ZT timepoints.

Frozen plant tissues of either Arabidopsis seedlings or tobacco leaves were homogenized in a reciprocal mixer mill (Retsch Mixer Mill MM 400, Newtown, Pennsylvania). Homogenized tissue of about 0.5 g was gently resuspended in 0.5 ml of SII buffer [100 mM sodium phosphate, pH 8.0, 150 mM NaCl, 5 mM EDTA, 5 mM EGTA, 0.1% Triton X-100, 1 mM PMSF, 1x protease inhibitor cocktail (Roche, Pleasanton, California), 1x Phosphatase Inhibitors II & III (Sigma), and 5 μM MG132 (Peptides International, Louisville, Kentucky)] and sonicated twice at 40% power, 1 s on/off cycles for a total of 10 s on ice (Fisher Scientific model FB505, with microtip probe, ThermoFisher Scientific, Waltham, Massachusetts). For PIF4p::PIF4-HA samples, about 100 μl homogenized tissue powder was mixed with 100 μl denature sample buffer (50 mM Tris-HCl, pH 7.5, 150 mM NaCl, 0.1% Triton X-100, 4% SDS) and denatured in dark by incubation at 95°C for 10 min. Extracts were then clarified by centrifugation twice at 4°C for 10 min at ≥20,000 g. For tobacco extracts, a 10% (w/v) of polyvinylpolypyrrolidone (PVPP) was added to resuspended extracts for 5 min incubation and was discarded after centrifugation. Concentration of total proteins from each sample was measured by using the DC Protein Assay kit (BIO-RAD). 40 ~ 50 μg total proteins were denatured and loaded to a 8% or 10% SDS-PAGE gel, followed by transferred to a nitrocellulose membrane.

For western blots, all of the following primary antibodies were diluted into PBS + 0.1% Tween + 2% BSA and incubated overnight at 4°C: Anti-GFP-rabbit (1:5000, Abcam, Cambridge, United Kingdom), anti-phyB-mouse (mAB2, at 1:3000, a generous gift from Dr. Akira Nagatani at Univeristy of Kyoto), and anti-ACTIN-mouse mAB1501 (1:2500, EMD-Millipore, Darmstadt, Germany). Anti-HA-HRP (Roche, Pleasanton, California) was used as 1:2000 and incubated for 1 hr. Anti-FLAGM2-HRP (Sigma Aldrich, St Louis, Missouri) and anti-RPT5-rabbit (ENZO Life Science, Farmingdale, New York) was incubated for 1 hr at room temperature and diluted into PBS + 0.1% Tween at 1:10,000 and 1:5000, respectively. Anti-Rabbit-HRP and anti-Mouse-HRP secondary antibodies (Sigma Aldrich, St Louis, Missouri) were diluted 1:20,000 into PBS + 0.1% Tween and incubated at room temperature for 1 hr.

## Co-immunoprecipitations (co-IPs) and in-vitro binding assay

For in vivo co-IP experiment, 2 mg of protein extract of PCH1ox3 plants (in 1 ml SII buffer with supplements of inhibitors) was used. Dynabeads (ThermoFisher Scientific, Waltham, Massachusetts) had been conjugated with the Anti-FLAGM2 monoclonal antibody (Sigma Aldrich, St Louis, Missouri) (*Nusinow et al., 2011*) to precipitate PCH1-His$_6$-FLAG$_3$ and its interacting proteins. 5 μg antibodies conjugated to 30 ul of Dynabeads were used for each FLAG-IP and were incubated with protein extracts on a rotor at 4°C for 1 hr, followed by being washed in SII buffer thrice. IP beads were added with 30 μl 2X SDS sample buffer and incubated at 75°C for 10 min to denature and elute bound proteins. SDS-PAGE and western detections were done as instructed above. It is noted that for co-IPs under different light treatments, all steps were carried out in a cold room supplemented with dim green safety light.

For in-vitro co-IP/binding assay, cDNAs of PCH1-His$_6$-Flag$_3$ or YPet-His$_6$-Flag$_3$ was gateway cloned into the pDEST17 vector (ThermoFisher Scientific, Waltham, Massachusetts). The fusion

proteins were expressed in BL21 (DE3) pLysS cells (Promega, Madison, Wisconsin) (1 mM IPTG induction for 3 hr at 30°C) and purified by Ni-NTA agarose beads (Qiagen, Hilden, Germany) following standard procedures. The phyB-HA prey was synthesized using plasmid pCMX-PL2-phyB-HA (*Qiu et al., 2015*) and the TNT T7 Quick Coupled Transcription/Translation System (Promega, Madison, Wisconsin) as instructed by manual. phyB-HA prey was first resuspended in 500 μl Tris-buffered saline (TBS) supplemented with 20 μM phytochromobilin (PΦB) and incubated for 45 min at 12°C under constant red (for Pfr phyB, 50 μmol·m$^{-2}$·s$^{-1}$), far red (for Pr phyB, 25 μmol·m$^{-2}$·s$^{-1}$) or dark (without PΦB, for phyB apoprotein) conditions. 4.9 ug purified PCH1-His$_6$-Flag$_3$ protein was then mixed with prey and incubated under the same light treatment for another 45 min at 12°C. 30 ul TALON beads (incubated for 30 min at 12°C) were used for immunoprecipitating each sample, followed by being washed with PBS+T buffer thrice.

### FLAG-His tandem affinity purification

Tandem affinity purifications using PCH1ox3 plants (in all genetic backgrounds) were carried out as previously described (*Huang et al., 2015*). In brief, 10-day-old seedlings of PCH1ox3 in Col, *elf4-2*, *elf3-2* and *phyB-9* genetic backgrounds were grown on sterilized qualitative filter paper, under the 12L:12D conditions. 5 g of whole seedlings were harvested at ZT12 and immediately frozen in liquid N$_2$. Tandem FLAG and His immunoprecipitations were carried out to co-purify proteins associated with PCH1-His$_6$-FLAG$_3$ as described in detail at Bio-protocol (*Huang and Nusinow, 2016*). At least two independent biological replications were performed.

### Protein digestion and identification using mass spectrometry

The proteins were cleaved to peptides with trypsin before analyzed on an LTQ-Orbitrap Velos Pro (ThermoFisher Scientific, Waltham, MA) coupled with a U3000 RSLCnano HPLC (Promega, Madison, Wisconsin) operated in positive ESI mode using collision induced dissociation (CID) to fragment the HPLC separated peptides as previously described (*Huang et al., 2015*).

### AP-MS data analysis

MS data were extracted by Proteome Discoverer (ThermoFisher Scientific; v.1.4) and database searches were done using Mascot (Matrix Science, London, UK; v.2.5.0) assuming the digestion enzyme trypsin, two missed cleavages, and using the TAIR10 database (20101214, 35,386 entries) and the cRAP database (http://www.thegpm.org/cRAP/). Deamidation of asparagine and glutamine, oxidation of methionine and carbamidomethyl of cysteine were specified as variable modifications, while a fragment ion mass tolerance of 0.80 Da, a parent ion tolerance of 15 ppm was used in the Mascot search. Scaffold (Proteome Software Inc., Portland, Oregon; v.4.4.3) was used to validate MS/MS based peptide and protein identifications. Peptide identifications were accepted if they could be established at greater than 95.0% probability and the Scaffold Local FDR was <1%. Protein identifications were accepted if they could be established at greater than 99.0% probability as assigned by the Protein Prophet algorithm (*Keller et al., 2002*; *Nesvizhskii et al., 2003*). Proteins that contained similar peptides and could not be differentiated based on MS/MS analysis alone were grouped to satisfy the principles of parsimony and proteins sharing significant peptide evidence were grouped into clusters. Only the proteins identified by PCH1ox3 AP-MS in Col with ≥2 unique peptides were presented in tables, except when proteins with only one peptide were identified in more than one replicate. A full list of all proteins co-purified by PCH1 AP-MS is in *Table 1—source data 1*. The mass spectrometry proteomics data have been deposited to the ProteomeXchange Consortium (*Vizcaino et al., 2014*) via the PRIDE partner repository with the dataset identifier PXD003352 and 10.6019/PXD003352.

## Acknowledgements

We thank Ivan Baxter, James C. Carrington, Elizabeth Haswell, Eirini Kaiserli and Takato Imaizumi for constructive comments. The phyB antibody was a gift from Akira Nagitani. Séverine Lorrain and Christian Fankhauser provided the PIF4-HA line. We acknowledge NSF (DBI-0922879) for acquisition of the LTQ-Velos Pro Orbitrap LC-MS/MS, (DBI-1337680) for acquisition of the Leica SP8-X confocal microscope, and NSF-REU (DBI-1156581) to Sona Pandey at the Danforth Center which supported JG.

# Additional information

## Funding

| Funder | Grant reference number | Author |
|---|---|---|
| National Science Foundation | IOS-1456796 | Dmitri Anton Nusinow |
| National Institute of General Medical Sciences | R01GM087388 | Meng Chen |

The funders had no role in study design, data collection and interpretation, or the decision to submit the work for publication.

## Author contributions

HH, Conception and design, Acquisition of data, Analysis and interpretation of data, Drafting or revising the article; CYY, RB, JG, AT, SA, Acquisition of data, Analysis and interpretation of data, Drafting or revising the article; MJN, Analysis and interpretation of data, Drafting or revising the article; BSE, Acquisition of data, Drafting or revising the article; MC, Conception and design, Acquisition of data, Analysis and interpretation of data; DAN, Conception and design, Analysis and interpretation of data, Drafting or revising the article

## Author ORCIDs

He Huang, http://orcid.org/0000-0002-4564-7604
Chan Yul Yoo, http://orcid.org/0000-0001-6159-7443
Rebecca Bindbeutel, http://orcid.org/0000-0001-7994-2073
Sophie Alvarez, http://orcid.org/0000-0001-8550-2832
Michael J Naldrett, http://orcid.org/0000-0002-6899-5652
Bradley S Evans, http://orcid.org/0000-0002-1207-9006
Dmitri A Nusinow, http://orcid.org/0000-0002-0497-1723

# Additional files

## Major datasets

The following datasets were generated:

| Author(s) | Year | Dataset title | Dataset URL | Database, license, and accessibility information |
|---|---|---|---|---|
| Huang H, Yoo CY, Bindbeutel R, Goldsworthy J, Tielking A, Alvarez S, Naldrett MJ, Evans BS, Chen M, Nusinow DA | 2015 | Tandem affinity purification and Mass spectrometry (TAP-MS) analysis of PCH1's associated proteins in various Arabidopsis genetic backgrounds. | http://proteomecentral.proteomexchange.org/dataset/PXD003352 | Publicly available at ProteomeXchange (accession no. PXD003352) |

The following previously published dataset was used:

| Author(s) | Year | Dataset title | Dataset URL | Database, license, and accessibility information |
|---|---|---|---|---|
| Huang H, Alvarez S, Bindbeutel R, Zhouxin S, Naldrett MJ, Evans BS, Briggs SP, Hicks LM, Kay SA, Nusinow DA | 2015 | Evening Complex associated proteins in Arabidopsis by AP-MS | http://proteomecentral.proteomexchange.org/dataset/PXD002606 | Publicly available at ProteomeXchange (accession no. PXD002606) |

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
