## [Decision Letter]

Thank you for submitting your work entitled "PCH1 integrates circadian and light-signaling pathways to control photoperiod-responsive growth in *Arabidopsis*" for consideration by *eLife*. Your article has been reviewed by two peer reviewers, and the evaluation has been overseen by a Reviewing Editor and Detlef Weigel as the Senior Editor. One of the two reviewers has agreed to share his identity: Julin Maloof.

The reviewers have discussed the reviews with one another and the Reviewing Editor has drafted this decision to help you prepare a revised submission.

Both reviewers as well as the Reviewing Editor are quite positive with respect to the quality of your study and in principle support its publication. Before final acceptance, we would however like to ask you to address the various comments, which you can find below, and submit an accordingly revised version.

*Reviewer #1:*

This paper described the characterization of a protein PCH1 which the authors show to associate with the Evening Complex (EC) and phyB to regulate hypocotyl elongation in short days. The manuscript is of particular interest because of how it integrates PCH1 into current models of light and circadian photoperiodic control of hypocotyl elongation. Notably PCH1 is shown to regulate the formation or stability of large phyB nuclear bodies, leading to the intriguing model that PCH1 is required to maintain phyB in its active pFR form after darkness. While this model is not explicitly tested (i.e. with in vitro measurements of phyB dark reversion) I think that the manuscript stands on its own as is.

The manuscript is clearly written, experiments are well documented, conclusions are justified.

*Reviewer #2:*

Huang et al. identify PCH1 as a novel interactor of active phyB that affects phyB photobody maintenance during night. Next to phytochromes, PCH1 also interacts with clock components, linking circadian clock and photoreceptor signaling. This is supported by strong affinity purification-MS data. PCH1 is a positive regulator of hypocotyl growth inhibition under short days, likely due to its repressive effect on PIF4 protein levels. The authors use a pch1 knock-out mutant, two complementation lines and two PCH1 overexpression lines to support their findings. The manuscript is well written, the data well documented and the findings are providing an important step forward in understanding the effect of photoperiod on plant growth. This is a strong manuscript and no additional experiments are requested from my side.

However, a few points need to be addressed:

1) Figure 4: Anti-GFP – It is unclear which part of the blot is shown for the FLAG-IP as no size markers are given. As phyB is tagged with GFP, phyB-GFP should be detected in the Co-IP with PCH1-FLAG (as detected by anti-phyB). The blot should be shown including the size range of phyB-GFP, otherwise it is misleading. Same for the input (now only bands when GFP alone is expressed, but phyB-GFP "not detected").

2) Figure 5 legend: "The percentage value indicates the mean% of all nuclei with the phenotypes shown (with or without photobodies).[…]". I think it is more appropriate to delete the "with the phenotypes shown" and indicate% consistently for "with photobodies"; i.e. to read "The percentage value indicates the mean% of all nuclei with photobodies.[…]" and accordingly change the percentages throughout in the cases where there are no photobodies (i.e. in pch1 phyB-9/PGB ZT 64 and ZT72 instead of giving percentage for "No PB" provide consistently percentage of with photobodies, i.e. 10% instead of 90%). E.g. in Figure 5—figure supplement 1: the phenotype of the nucleus shown in "ox3" line ZT56 is clearly many more nuclear photobodies than in wild type (phyB-9/PGB). However, the 100% is misleading, as definitely not all nuclei show this phenotype (see quantification).

Figure 5—figure supplement 2: Isn't the nucleus shown for PCH1ox3 phyB-9/PGB showing small photobodies? I.e. does percentage indicate "all nuclei without photobodies" (as in legend) or "all nuclei without large photobodies" (as suggested by the title of the figure)?

3) Figure 6: Provide the actual etiolated hypocotyl lengths of the 5 genotypes in the figure legend.

4) In the subsection “*pch1* causes defects specifically in red light and up-regulates expression of downstream transcription factors”, the authors state: "These data show that PCH1 specifically modulates hypocotyl length in response to red light". That is a too strong conclusion, as particularly the FR data seems to be a saturated response. Figure 6—figure supplement 1: WT is 1mm under these conditions; cannot be shorter. "Data show" should be changed to "data suggest", or a dose-response analysis should be shown.

---

## [Author Response]

Reviewer #2:

[…] However, a few points need to be addressed:

*1) Figure 4: Anti-GFP – It is unclear which part of the blot is shown for the FLAG-IP as no size markers are given. As phyB is tagged with GFP, phyB-GFP should be detected in the Co-IP with PCH1-FLAG (as detected by anti-phyB). The blot should be shown including the size range of phyB-GFP, otherwise it is misleading. Same for the input (now only bands when GFP alone is expressed, but phyB-GFP "not detected").*

The nitrocellulose membrane containing all the inputs and IP lanes were first cut into three strips based on the molecular weights of the target proteins; from 100kD to 250kD for phyB-GFP, from 50kD to 100kD for PCH1-FLAG, and from 25kD to 50kD for GFP. These strips of membrane were separately detected with anti-phyB, anti-FLAG or anti-GFP, respectively. In order to reduce confusion, the authors rearranged panels according to protein size and included the position of molecular weight markers.

*2) Figure 5 legend: "The percentage value indicates the mean% of all nuclei with the phenotypes shown (with or without photobodies).[…]". I think it is more appropriate to delete the "with the phenotypes shown" and indicate% consistently for "with photobodies"; i.e. to read "The percentage value indicates the mean% of all nuclei with photobodies.[…]" and accordingly change the percentages throughout in the cases where there are no photobodies (i.e. in pch1 phyB-9/PGB ZT 64 and ZT72 instead of giving percentage for "No PB" provide consistently percentage of with photobodies, i.e. 10% instead of 90%). E.g. in Figure 5—figure supplement 1: the phenotype of the nucleus shown in "ox3" line ZT56 is clearly many more nuclear photobodies than in wild type (phyB-9/PGB). However, the 100% is misleading, as definitely not all nuclei show this phenotype (see quantification).*

For clarity, the authors felt that the representative image shown should represent the majority of observations ± photobodies, and therefore the last time points should show no photobody. Of the nuclei with photobodies, we then go on to count the number and size distribution of the photobodies within the nuclei. In Figure 5—figure supplement 1, the ox3 line has more photobodies, compared to WT, reflecting the representative image shown. With that said, we revised the legends to make it easier to be understood:

“Representative confocal images showing phyB-GFP-containing photobodies in *phyB-9, pch1phyB-9* and PCH1ox3 *phyB-9* plants at indicated time points during light-to-dark transition. […] The percentage of nuclei showing the corresponding photobody patterns (with or without photobodies) were calculated based on three independent experiments. N represents the total number of nuclei analyzed for each time point. Scale bars equal to 5 μm.”

*Figure 5—figure supplement 2: Isn't the nucleus shown for PCH1ox3 phyB-9/PGB showing small photobodies? I.e. does percentage indicate "all nuclei without photobodies" (as in legend) or "all nuclei without large photobodies" (as suggested by the title of the figure)?*

Because only a small amount of phyB-GFP are localized in the nucleus in the dark, the images of nuclei taken in true dark conditions are not of sufficient contrast for robust automated photobody counting and size determination as was performed for the other quantified images (Figure 5 and Figure 5—figure supplement 1), therefore the authors felt that the most conservative action is to remove this supplementary figure and the part of the sentence in the Discussion that refers to this figure.

*3) Figure 6: Provide the actual etiolated hypocotyl lengths of the 5 genotypes in the figure legend.*

The authors included the etiolated hypocotyl lengths of the 5 genotypes in the figure legend as mean ± SD for comparison.

4) In the subsection “pch1 causes defects specifically in red light and up-regulates expression of downstream transcription factors”, the authors state: "These data show that PCH1 specifically modulates hypocotyl length in response to red light". That is a too strong conclusion, as particularly the FR data seems to be a saturated response. Figure 6—figure supplement 1: WT is 1mm under these conditions; cannot be shorter. "Data show" should be changed to "data suggest", or a dose-response analysis should be shown.

The authors have changed the sentence from “show” to “suggest”.